# PEARL: Data Synthesis via Private Embeddings and Adversarial Reconstruction Learning

**Seng Pei Liew, Tsubasa Takahashi, Michihiko Ueno**
LINE Corporation
`{sengpei.liew,tsubasa.takahashi,michihiko.ueno}@linecorp.com`

## Abstract

We propose a new framework of synthesizing data using deep generative models in a differentially private manner. Within our framework, sensitive data are sanitized with rigorous privacy guarantees in a one-shot fashion, such that training deep generative models is possible without re-using the original data. Hence, no extra privacy costs or model constraints are incurred, in contrast to popular gradient sanitization approaches, which, among other issues, cause degradation in privacy guarantees as the training iteration increases. We demonstrate a realization of our framework by making use of the characteristic function and an adversarial re-weighting objective, which are of independent interest as well. Our proposal has theoretical guarantees of performance, and empirical evaluations on multiple datasets show that our approach outperforms other methods at reasonable levels of privacy.

## 1 Introduction

Synthesizing data under differential privacy (DP) (Dwork (2006; 2011); Dwork et al. (2014)) enables us to share the synthetic data and generative model with rigorous privacy guarantees. Particularly, DP approaches of data synthesis involving the use of deep generative models have received attention lately (Takagi et al. (2021); Xie et al. (2018); Torkzadehmahani et al. (2019); Frigerio et al. (2019); Jordon et al. (2019); Chen et al. (2020); Harder et al. (2021)).

Typically, the training of such models utilizes gradient sanitization techniques (Abadi et al. (2016)) that add noises to the gradient updates to preserve privacy. While such methods are conducive to deep learning, due to composability, each access to data leads to degradation in privacy guarantees, and as a result, the training iteration is limited by the privacy budget. Recently, Harder et al. (2021) has proposed DP-MERF, which first represents the sensitive data as random features in a DP manner and then learns a generator by minimizing the discrepancy between the (fixed) representation and generated data points. DP-MERF can iterate the learning process of the generator without further consuming the privacy budget; however, it is limited in the learning and generalization capabilities due to its fixed representation. *In this work, we seek a strategy of training deep generative models privately that is able to resolve the aforementioned shortcomings, and is practical in terms of privacy (e.g., usable image data at $\epsilon \simeq 1$.)*

We propose a private learning framework called **PEARL** (Private Embeddings and Adversarial Reconstruction Learning). In this framework, we have i) *no limitation in learning iterations*, and ii) *well-reconstruction capability*. Towards those preferable properties, our framework first obtains (1) informative embedding of sensitive data and (2) auxiliary information (e.g., hyperparameters) useful for training, both in a differentially private manner, then (3) the generative model is trained implicitly like GANs via the private embedding and auiliary information, where the learning is based on a stochastic procedure that generates data, and (4) a critic distinguishing between the real and generated data. The overview of PEARL is illustrated in Fig. 1.

As a concrete realization of PEARL, We first identify that the characteristic function (CF) representation of data can be sanitized as the private embedding of PEARL. Consequently, it is possible to train deep generative models using an appropriately defined metric measuring the discrepancy between the real (but sanitized) and generated data distribution based on the CF without re-using the original data. As will be explained in detail in later Sections, the generative modelling approach using CFs also

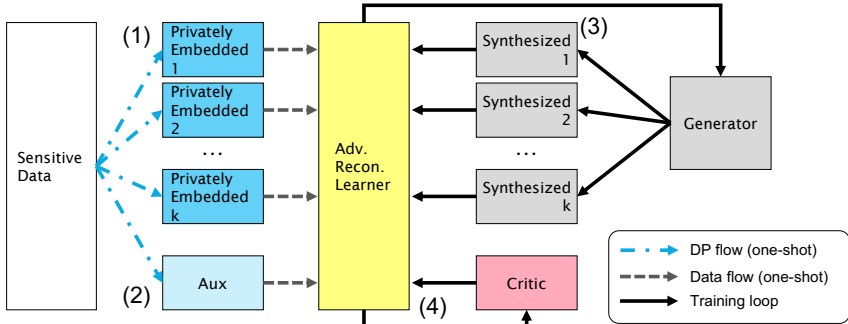

Figure 1: PEARL is a private learning framework that has i) no limitation in learning iterations, and ii) well-reconstruction capability. Towards those preferable properties, our framework first obtains (1) private embedding and (2) auxiliary information from the sensitive data, then (3) trains a generator while (4) optimizing a critic to distinguish between the real and generated data.

involves sampling "frequencies" from an *ad hoc* distribution, to project the data to the embedding. It is desirable to optimize the sampling distribution to better represent the data as an embedding, but the naive way of optimizing it would require re-accessing the data via sampling, coming at a cost of privacy budget. Henceforth, we also propose to incorporate a privacy-preserving critic to optimize the sampling strategy, which, through re-weighting, chooses the best representation from a fixed samples of frequencies without extra privacy costs.

To this end, we propose the following minimax optimization training objective:

$$\inf_{\theta \in \Theta} \sup_{\omega \in \Omega} \sum_{i=1}^{k} \frac{\omega(\mathbf{t}_i)}{\omega_0(\mathbf{t}_i)} \big| \widetilde{\Phi}_{\mathbb{P}_r}(\mathbf{t}_i) - \widehat{\Phi}_{\mathbb{Q}_\theta}(\mathbf{t}_i) \big|^2. \tag{1}$$

See later parts for notations and details. Theoretically, we show that our proposed objective has properties similar to those that are suited to training GANs, i.e., continuity and differentiability of the generator's parameters, and continuity in weak topology. We also prove the consistency of our privacy-preserving sampling strategy at the asymptotic limit of infinite sampling. Empirical evaluations show that PEARL is able to high-quality synthetic data at reasonable privacy levels.

**Related works.** Traditional methods of synthesizing data are mainly concerned with discrete data or data preprocessed to the discrete form (Zhang et al. (2017); Qardaji et al. (2014); He et al. (2015); Chen et al. (2015); Cai et al. (2021); Zhang et al. (2021)), whereas we are interested in more general methods involving continuous data. Deep generative models under the DP setting are suitable for this type of tasks (Takagi et al. (2021); Xie et al. (2018); Torkzadehmahani et al. (2019); Frigerio et al. (2019); Jordon et al. (2019); Chen et al. (2020); Harder et al. (2021)). The private training of deep generative models is usually performed using gradient sanitization methods. An exception is DP-MERF (Harder et al. (2021)), which is closest to our work. There, random features used to approximate the maximum mean discrepancy (MMD) objective are privatized and utilized for training a generator. PEARL, which, as a realization, uses CFs, may be viewed as a generalization of DP-MERF. Additionally, PEARL has several distinctive features which are lacking in DP-MERF. The first lies in the introduction of a privacy-preserving critic, which leads to an improvement of performance. The second is the private selection of the parameter of the sampling distribution, which is also shown to be vital. Moreover, DP-MERF uses non-characteristic kernels when treating tabular data, in contrast to ours, which is characteristic and has guarantees in convergence. We finally note that generative models using CFs but only non-privately have been explored before (Ansari et al. (2020); Li et al. (2020)) .

**Contributions.** Our contribution in this paper is three-fold: (i) We propose a general framework called PEARL, where, unlike gradient sanitization methods, the generator training process and iteration are unconstrained; reliance on ad-hoc (non-private) hyperparameter tuning is reduced by extracting hyperparameters (auxiliary information) privately. (ii) We demonstrate a realization of our framework by making use of the characteristic function and an adversarial re-weighting objective. (iii) Our proposal has theoretical guarantees of performance, and empirical evaluations show that our approach outperforms competitors at reasonable levels of privacy ($\epsilon \simeq 1$).

## 2 PRELIMINARIES

This Section gives a brief review of essential preliminaries about differential privacy, characteristic function and the related notations.

### 2.1 DIFFERENTIAL PRIVACY

**Definition 1** (($\epsilon, \delta$)-Differential Privacy). *Given privacy parameters $\epsilon \geq 0$ and $\delta \geq 0$, a randomized mechanism, $\mathcal{M} : \mathcal{D} \to \mathcal{R}$ with domain $\mathcal{D}$ and range $\mathcal{R}$ satisfies ($\epsilon, \delta$)-differential privacy (DP) if for any two adjacent inputs $d, d' \in \mathcal{D}$ and for any subset of outputs $\mathcal{S} \subseteq \mathcal{R}$, the following holds:*

$$\Pr[\mathcal{M}(d) \in S] \leq e^{\epsilon} \cdot \Pr[\mathcal{M}(d') \in S] + \delta. \tag{2}$$

We next consider concrete ways of sanitizing certain outputs with DP. A typical paradigm of DP is applying the randomized mechanism, $\mathcal{M}$, to a certain deterministic function $f : \mathcal{D} \to \mathbb{R}$ such that the output of $f$ is DP. The noise magnitude added by $\mathcal{M}$ is determined by the *sensitivity* of $f$, defined as $\Delta_f = \sup_{d, d' \in \mathcal{D}} \|f(d) - f(d')\|$, where $\|\cdot\|$ is a norm function defined on $f$'s output domain. $d$ and $d'$ are any adjacent pairs of dataset. Laplacian and Gaussian mechanisms are the standard randomized mechanisms. We primarily utilize the Gaussian mechanism in this paper (Dwork et al. (2014)):

**Definition 2** (Gaussian Mechanism). *Let $f : X \to \mathbb{R}$ be an arbitrary function with sensitivity $\Delta_f$. The Gaussian Mechanism, $\mathcal{M}_\sigma$, parameterized by $\sigma$, adds noise to the output of $f$ as follows:*

$$\mathcal{M}_\sigma(x) = f(x) + \mathcal{N}(0, \sigma^2 I). \tag{3}$$

One of the most important properties of DP relevant to our work is the post-processing theorem (Dwork et al. (2014)):

**Theorem 1** (Post-processing Theorem). *Let $\mathcal{M} : \mathcal{D} \to \mathcal{R}$ be ($\epsilon, \delta$)-DP and let $f : \mathcal{R} \to \mathcal{R}'$ be an arbitrary randomized function. Then, $f \circ \mathcal{M} : \mathcal{D} \to \mathcal{R}'$ is ($\epsilon, \delta$)-DP.*

It ensures that the DP-sanitized data can be re-used without further consuming privacy budgets.

### 2.2 CHARACTERISTIC FUNCTIONS

Characteristic function (CF) is widely utilized in statistics and probability theory, and perhaps is best known to be used to prove the central limit theorem (Williams (1991)). The definition is as follows.

**Definition 3** (Characteristic Function). *Given a random variable $X \subseteq \mathbb{R}^d$ and $\mathbb{P}$ as the probability measure associated with it, and $\mathbf{t} \in \mathbb{R}^d$, the corresponding characteristic function (CF) is given by*

$$\Phi_{\mathbb{P}}(\mathbf{t}) = \mathbb{E}_{\mathbf{x} \sim \mathbb{P}}[e^{i\mathbf{t} \cdot \mathbf{x}}] = \int_{\mathbb{R}^d} e^{i\mathbf{t} \cdot \mathbf{x}} d\mathbb{P}. \tag{4}$$

Here, $i$ is the imaginary number. From the signal processing point of view, this mathematical operation is equivalent to the Fourier transformation, and $\Phi_{\mathbb{P}}(\mathbf{t})$ is the Fourier transform at frequency $\mathbf{t}$. It is noted that we deal with the discrete approximation of CFs in practice. That is, given a dataset with $n$ i.i.d. samples, $\{\mathbf{x}_j\}_{j=1}^n$ from $\mathbb{P}$, the empirical CF is written as $\widehat{\Phi}_{\mathbb{P}}(\mathbf{t}) = \frac{1}{n} \sum_{i=j}^n e^{i\mathbf{t} \cdot \mathbf{x}_j}$. We next introduce characteristic function distance (CFD) (Heathcote (1972); Chwialkowski et al. (2015)):

**Definition 4** (Characteristic Function Distance). *Given two distributions $\mathbb{P}$ and $\mathbb{Q}$ of random variables residing in $\mathbb{R}^d$, and $\omega$ a sampling distribution on $\mathbf{t} \in \mathbb{R}^d$, the squared characteristic function distance (CFD) between $\mathbb{P}$ and $\mathbb{Q}$ is computed as:*

$$\mathcal{C}^2(\mathbb{P}, \mathbb{Q}) = \mathbb{E}_{\mathbf{t} \sim \omega(\mathbf{t})} \left[ \left| \Phi_{\mathbb{P}}(\mathbf{t}) - \Phi_{\mathbb{Q}}(\mathbf{t}) \right|^2 \right] = \int_{\mathbb{R}^d} \left| \Phi_{\mathbb{P}}(\mathbf{t}) - \Phi_{\mathbb{Q}}(\mathbf{t}) \right|^2 \omega(\mathbf{t}) d\mathbf{t}. \tag{5}$$

**Notations.** Let us make a short note on the notations before continuing. Let $k$ be the number of $\mathbf{t}$ drawn from $\omega$ and $\mathbb{P}$ be the probability measure of a random variable. We group the CFs associated to $\mathbb{P}$ of different frequencies, $(\widehat{\Phi}_{\mathbb{P}}(\mathbf{t}_1), \dots, \widehat{\Phi}_{\mathbb{P}}(\mathbf{t}_k))^\top$ more compactly as $\widehat{\boldsymbol{\phi}}_{\mathbb{P}}(\mathbf{x})$. To make the dependence of $\widehat{\boldsymbol{\phi}}_{\mathbb{P}}(\mathbf{x})$ on the sampled data explicit, we also use the following notation: $\widehat{\boldsymbol{\phi}}_{\mathbb{P}}(\mathbf{x}) = \frac{1}{n} \sum_{j=1}^n \widehat{\boldsymbol{\phi}}_{\mathbb{P}}(\mathbf{x}_j)$. We notice that $\|\widehat{\boldsymbol{\phi}}_{\mathbb{P}}(\mathbf{x})\|_2 \equiv \sqrt{\sum_{m=1}^k |\widehat{\Phi}_{\mathbb{P}}(\mathbf{t}_m)|^2} = \sqrt{\sum_{m=1}^k |\sum_{l=1}^n e^{i\mathbf{t}_m \cdot \mathbf{x}_l}/n|^2} \leq \sqrt{\sum_{m=1}^k |\sum_{l=1}^n 1/n|^2} = \sqrt{k}$, where the norm is taken over the (complex) frequency space. With a slight abuse of notation, we abbreviate $\widehat{\boldsymbol{\phi}}_{\mathbb{P}}$ as $\widehat{\boldsymbol{\phi}}$ when there is no ambiguity in the underlying probability measure associated with the CF.

## 3    GENERATIVE MODEL OF PEARL

Let us describe generative modelling under the PEARL framework. The realization is summarized by the following Proposition:

**Proposition 1.** *Let the real data distribution be $\mathbb{P}_r$, and the output distribution of an implicit generative model $G_\theta$ by $\mathbb{Q}_\theta$. Let $n$ be the total number of real data instances and consider releasing $k$ CFs from the dataset. Then, $G_\theta$ trained to optimize the empirical CFD, $\min_{\theta \in \Theta} \widehat{\mathcal{C}}^2(\mathbb{P}_r, \mathbb{Q}_\theta)$ with the CF sanitized according to the Gaussian mechanism (Defn. 3) with sensitivity $2\sqrt{k}/n$ satisfies $(\epsilon, \delta)$-DP, where $\sigma \geq \sqrt{2 \log(1.25/\delta)}/\epsilon$.*

*Proof.* In the following we give a detailed explanation of the above Proposition. The first step of PEARL is projecting the data to the CF as in Eq. 4, where the number of embeddings is the number of frequency drawn from $\omega(\mathbf{t})$.

We note that CF has several attractive properties. CF is uniformly continuous and bounded, as can be seen from its expression in Eq. 4. Unlike the density function, the CF of a random variable always exists, and the uniqueness theorem implies that two distributions are identical if and only if the CFs of the random variables are equal (Lukacs (1972)).

The CF is sanitized with DP by applying the Gaussian mechanism (Defn. 3) to $\widehat{\phi}(\mathbf{x})$:

$$\widetilde{\phi}(\mathbf{x}) = \widehat{\phi}(\mathbf{x}) + \mathcal{N}(0, \Delta_{\widehat{\phi}(\mathbf{x})}^2 \sigma^2 I), \tag{6}$$

where we write the sanitized CF as $\widetilde{\phi}(\mathbf{x})$; $\Delta_{\widehat{\phi}(\mathbf{x})}$ denotes the sensitivity of the CF, $\sigma$ denotes the noise scale which is determined by the privacy budget, $(\epsilon, \delta)$.

The calculation of sensitivity is tractable (no ad-hoc clipping commonly required by gradient sanitization methods) Without loss of generality, consider two neighboring datasets of size $n$ where only the last instance differs ($\mathbf{x}_n \neq \mathbf{x}'_n$). The sensitivity of $\widehat{\phi}(\mathbf{x})$ may then be calculated as

$$\Delta_{\widehat{\phi}(\mathbf{x})} = \max_{\mathcal{D}, \mathcal{D}'} \left\| \tfrac{1}{n} \sum_{j=1}^{n} \widehat{\phi}(\mathbf{x}_j) - \tfrac{1}{n} \sum_{j=1}^{n} \widehat{\phi}(\mathbf{x}'_j) \right\|_2 = \max_{\mathbf{x}_n, \mathbf{x}'_n} \left\| \tfrac{1}{n} \widehat{\phi}(\mathbf{x}_n) - \tfrac{1}{n} \widehat{\phi}(\mathbf{x}'_n) \right\|_2 = \tfrac{2\sqrt{k}}{n},$$

where we have used triangle inequality and $\|\widehat{\phi}(\cdot)\|_2 \leq \sqrt{k}$. It can be seen that the sensitivity is proportional to the square root of the number of embeddings (releasing more embeddings requires more DP noises to be added to $\widehat{\phi}$) but inversely proportional to the dataset size, which is important for controlling the magnitude of noise injection at practical privacy levels, as will be discussed in later Sections.

We would like to train a generative model where the CFs constructed from the generated data distribution, $\mathcal{Y} \subseteq \mathbb{R}^d$, matches those (sanitized) from the real data distribution, $\mathcal{X} \subseteq \mathbb{R}^d$. A natural way of achieving this is via implicit generative modelling (MacKay (1995); Goodfellow et al. (2014)). We introduce a generative model parametrized by $\theta$, $G_\theta : \mathcal{Z} \to \mathbb{R}^d$, which takes a low-dimensional latent vector $z \in \mathcal{Z}$ sampled from a pre-determined distribution (e.g., Gaussian distribution) as the input.

In order to quantify the discrepancy between the real and generated data distribution, we use the CFD defined in Eq. 5. Empirically, when a finite number of frequencies, $k$, are sampled from $\omega$, $\mathcal{C}^2(\mathbb{P}, \mathbb{Q})$ is approximated by

$$\widehat{\mathcal{C}}^2(\mathbb{P}, \mathbb{Q}) = \frac{1}{k} \sum_{i=1}^{k} \left| \widehat{\Phi}_{\mathbb{P}}(\mathbf{t}_i) - \widehat{\Phi}_{\mathbb{Q}}(\mathbf{t}_i) \right|^2 \equiv \left\| \widehat{\phi}_{\mathbb{P}}(\mathbf{x}) - \widehat{\phi}_{\mathbb{Q}}(\mathbf{x}) \right\|_2^2, \tag{7}$$

where $\widehat{\Phi}_{\mathbb{P}}(\mathbf{t})$ and $\widehat{\Phi}_{\mathbb{Q}}(\mathbf{t})$ are the empirical CFs evaluated from i.i.d. samples of distributions $\mathbb{P}$ and $\mathbb{Q}$ respectively. The training objective of the generator is to find the optimal $\theta \in \Theta$ that minimizes the empirical CFD: $\min_{\theta \in \Theta} \widehat{\mathcal{C}}^2(\mathbb{P}_r, \mathbb{Q}_\theta)$. It can be shown via uniqueness theorem that as long as $\omega$ resides in $\mathbb{R}^d$, $\mathcal{C}(\mathbb{P}, \mathbb{Q}) = 0 \iff \mathbb{P} = \mathbb{Q}$ (Sriperumbudur et al. (2011)). This makes CFD an ideal distance metric for training the generator.

**Optimization procedure.** The generator parameter, $\theta$, is updated as follows. $\widehat{\phi}_{\mathbb{P}_r}(\mathbf{x})$ is first sanitized to obtain $\widetilde{\phi}_{\mathbb{P}_r}(\mathbf{x})$, as in Eq. 6. This is performed only for once (one-shot). Then, at each iteration, $m$ samples of $\mathbf{z}$ are drawn to calculate $\widehat{\phi}_{\mathbb{Q}_\theta}(\mathbf{x})$. Gradient updates on $\theta$ are performed by minimizing the CFD, $\left\|\widetilde{\phi}_{\mathbb{P}_r}(\mathbf{x}) - \widehat{\phi}_{\mathbb{Q}_\theta}(\mathbf{x})\right\|_2$.

We note that only the first term, $\widetilde{\phi}_{\mathbb{P}_r}(\mathbf{x})$, has access to the real data distribution, $\mathcal{X}$, of which privacy is of concern. Then, by Thm. 1, $G_\theta$ trained with respect to $\widetilde{\phi}_{\mathbb{P}_r}(\mathbf{x})$ is DP. Furthermore, unlike gradient sanitization methods, the training of $G_\theta$ is not affected by network structure or training iterations. Once the sanitized CF is released, there is no additional constraints due to privacy on the training procedure. $\qquad\square$

**DP release of auxiliary information.** Auxiliary information, e.g., hyperparameters regarding to the dataset useful for generating data with better quality may be obtained under DP. In our case, one may extract auxiliary information privately from the dataset to select good parameters for the sampling distribution $\omega(\mathbf{t})$. This will be discussed in more detail in the next Sections.

Another example is the modelling of tabular data. Continuous columns of tabular data consisting of multiple modes may be modelled using Gaussian mixture models (GMMs) (Xu et al. (2019)). GMMs are trainable under DP using a DP version of the expectation-maximization algorithm (Park et al. (2017)). The total privacy budget due to multiple releases of information is accounted for using the Rényi DP composition. See App. A for the definition of Rényi DP.

## 4    Adversarial Reconstruction Learning

This Section is devoted to proposing a privacy-preserving critic for optimizing $\omega(\mathbf{t})$, while giving provable guarantees of performance.

Back to Eq. 5 and Eq. 7, we note that choosing a "good" $\omega(\mathbf{t})$ or a "good" set of sampled frequencies is vital at helping to discriminate between $\mathbb{P}$ and $\mathbb{Q}$. For example, if the difference between $\mathbb{P}$ and $\mathbb{Q}$ lies in the high-frequency region, one should choose to use $\mathbf{t}$ with large values to, in the language of two-sample testing, improve the test power.

**Adversarial objective.** If the resulting empirical CFD remains small due to under-optimized $\omega(\mathbf{t})$, while the two distributions still differ significantly, the generator cannot be optimally trained to generate high-quality samples resembling the real data. Hence, we consider training the generator by, in addition to minimizing the empirical CFD, maximizing the empirical CFD using an *adversarial objective* which acts as a *critic*, where the empirical CFD is maximized by finding the best sampling distribution. We consider a training objective of the form

$$\inf_{\theta \in \Theta} \sup_{\omega \in \Omega} \widehat{\mathcal{C}_\omega}^2(\mathbb{P}_r, \mathbb{Q}_\theta), \tag{8}$$

where $\Omega$ is some set of probability distribution space of which the sampling distribution $\omega$ lives in.

**Privacy-preserving optimization.** It is intractable to directly optimize $\omega$ in the integral form as in Eq. 5. In this work, we opt for a *privacy-preserving one-shot* sampling strategy where once the private data is released (by sampling from $\omega$), optimization is performed without further spending the privacy budget. We believe that such a new formulation of distribution learning is of independent interest as well.

**Effectiveness of the critic.** To further motivate why it is preferable to introduce an adversarial objective as in Eq. 8, we present a simple demonstration through the lens of two-sample testing (Chwialkowski et al. (2015); Jitkrittum et al. (2016)) using synthetic data generated from Gaussian distributions. We generate two unit-variance multivariate Gaussian distributions $\mathbb{P}, \mathbb{Q}$, where all dimensions but one have zero mean. We conduct a two-sample test using the CF to distinguish between the two distributions, which gets more difficult as the dimensionality increases. We test if the null hypothesis where the samples are drawn from the same distribution is rejected. Higher rejection rate indicates better test power.

Note that the first dimension (where the distribution differs) of the frequency used to construct CFs is the most discriminative dimension for distinguishing the distributions. We consider three sets of

frequencies: "unoptimized", "normal", and "optimized", where the set of "unoptimized" frequencies is optimized with re-weighting. More details can be found in App. C.

Fig. 2 shows the hypothesis rejection rate versus the number of dimensions for the three cases considered above. As can be observed, the "optimized" case gives overall the best test power. While this experiment is somewhat contrived, it can be understood that although both "unoptimized" and "optimized" sets of frequencies contain the discriminative $\mathbf{t}_0$, the re-weighting procedure of selecting the most discriminative CF improves the test power significantly. Even without re-sampling from a "better" $\omega(\mathbf{t})$, re-weighting the existing frequencies can improve the test power. The fact that re-weighting can improve the test power is crucial privacy-wise because the altenative method, re-sampling, causes degradation in privacy.

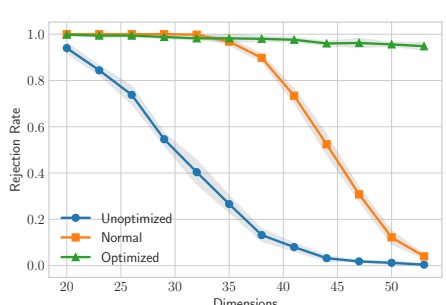

Figure 2: Increased test power upon optimization (green) in two-sample test.

**Proposal.** Recall that the empirical CFD, $\widehat{\mathcal{C}}^2(\mathbb{P}_r, \mathbb{Q}_\theta) = \frac{1}{k}\sum_{i=1}^{k}\left|\widehat{\Phi}_{\mathbb{P}_r}(\mathbf{t}_i) - \widehat{\Phi}_{\mathbb{Q}_\theta}(\mathbf{t}_i)\right|^2$, is obtained by drawing $k$ frequencies from a base distribution $\omega_0$. Our idea is to find a (weighted) set of frequencies that gives the best test power from the drawn set. We propose Eq. 1 as the optimization objective, restated below:

$$\inf_{\theta \in \Theta} \sup_{\omega \in \Omega} \sum_{i=1}^{k} \frac{\omega(\mathbf{t}_i)}{\omega_0(\mathbf{t}_i)}\left|\widetilde{\Phi}_{\mathbb{P}_r}(\mathbf{t}_i) - \widehat{\Phi}_{\mathbb{Q}_\theta}(\mathbf{t}_i)\right|^2.$$

Note that the generator trained with this objective still satisfies DP as given in Prop. 1 due to Thm. 1. The following Lemma ensures that the discrete approximation of the inner maximization of Eq. 1 approaches the population optimum as the number of sampling frequency increases ($k \rightarrow \infty$):

**Lemma 1.** *Let $\omega_0$ be any probability distribution defined on $\mathcal{R}^d$, and let $f : \mathcal{R}^d \rightarrow \mathcal{R}'$ be any function. Also let $\mathbf{t} \in \mathcal{R}^d$ and $\omega^*$ be the maximal distribution of $\omega$ with respect to $\mathbb{E}_\omega[f(\mathbf{t})] \equiv \int f(\mathbf{t})\omega(\mathbf{t})d\mathbf{t}$. Assume that the empirical approximation $\widehat{\mathbb{E}}_\omega[f(\mathbf{t})] \rightarrow \mathbb{E}_\omega[f(\mathbf{t})]$ at the asymptotic limit for any $\omega$. Then, $\widehat{\mathbb{E}}_{\omega_0}[f(\mathbf{t})\frac{\omega^*(\mathbf{t})}{\omega_0(\mathbf{t})}] \rightarrow \mathbb{E}_{\omega^*}[f(\mathbf{t})]$ at the asymptotic limit as well.*

The proof is in App. B, and is based on importance sampling. Empirically, we find that treating $\omega(\mathbf{t}_i)$ as a free parameter and optimizing it directly does not lead to improvement in performance. This may be due to the optimization procedure focusing too much on uninformative frequencies that contain merely noises due to DP or sampling. We perform *parametric* optimization instead, that is, e.g., we perform optimization with respect to $\{\boldsymbol{\mu}, \boldsymbol{\sigma}\}$ if $\omega$ is of the Gaussian form, $\mathcal{N}(\boldsymbol{\mu}, \boldsymbol{\sigma}^2)$.

**Performance guarantees.** Let us discuss the theoretical properties of Eq. 8. The objective defined in Eq. 8 shares beneficial properties similar to those required to train good GANs, first formulated in (Arjovsky et al. (2017)). First, the generator learns from a distance continuous and differentiable almost everywhere within the generator's parameters. Second, the distance is continuous in weak topology, and thus provides informative feedback to the generator (different from, e.g., the Jensen-Shannon divergence, which does not satisfy this property). We make assumptions similar to those given in (Arjovsky et al. (2017)), and state the first theorem as follows.

**Theorem 2.** *Assume that $G_\theta(z)$ is locally Lipschitz with respect to $(\theta, z)$; there exists $L((\theta, z)$ satisfying $\mathbb{E}_z[L(\theta, z)] < \infty$; and $\sup_{\omega \in \Omega}\mathbb{E}_\omega[|\mathbf{t}|] < \infty$ for all $\mathbf{t}$. Then, the function $\sup_{\omega \in \Omega}\mathcal{C}^2_\omega(\mathbb{P}_r, \mathbb{Q}_\theta)$ is continuous in $\theta \in \Theta$ everywhere, and differentiable in $\theta \in \Theta$ almost everywhere.*

Note that the local Lipschitz assumptions are satisfied by commonly used neural network components, such as fully connected layers and ReLU activations. The continuity and differentiability conditions with respect to $\theta$ stated above allow $G_\theta$ to be trained via gradient descent. The theorem related to continuity in weak topology is the following:

**Theorem 3.** *Let $\mathbb{P}$ be a distribution on $\mathcal{X}$ and $(\mathbb{P}_n)_{n \in \mathbb{N}}$ be a sequence of distributions on $\mathcal{X}$. Under the assumption $\sup_{\omega \in \Omega}\mathbb{E}_{\omega(\mathbf{t})}[\|\mathbf{t}\|] < \infty$, the function $\sup_{\omega \in \Omega}\mathcal{C}^2_\omega(\mathbb{P}_n, \mathbb{P})$ is continuous in the*

*weak topology, i.e., if* $\mathbb{P}_n \xrightarrow{D} \mathbb{P}$, *then* $\sup_{\omega \in \Omega} \mathcal{C}_\omega^2(\mathbb{P}_n, \mathbb{P}) \xrightarrow{D} 0$, *where* $\xrightarrow{D}$ *denotes convergence in distribution.*

Weakness is desirable as the easier (weaker) it is for the distributions to converge, the easier it will be for the model to learn to fit the data distribution. The core ideas used to prove both of these theorems are the fact that the difference of the CFs (which is of the form $e^{ia}$) can be bounded as follows: $|e^{ia} - e^{ib}| \leq |a - b|$, and showing that the function is locally Lipschitz, which ensures the desired properties of continuity and differentiability. See App. B for the full proofs.

## 5 EXPERIMENTS

### 5.1 EXPERIMENTAL SETUP

To test the efficacy of PEARL, we perform empirical evaluations on three datasets, namely MNIST (LeCun et al. (2010)), Fashion-MNIST (Xiao et al. (2017)) and Adult (Asuncion & Newman (2007)). Detailed setups are available in App. H.

**Training Procedure.** As our training involves minimax optimization (Eq. 1), we perform gradient descent updates based on the minimization and maximization objectives alternately. We use a zero-mean diagonal standard-deviation Gaussian distribution, $\mathcal{N}(\mathbf{0}, \mathrm{diag}(\boldsymbol{\sigma}^2)$ as the sampling distribution, $\omega$. Maximization is performed with respect to $\boldsymbol{\sigma}$. Let us give a high-level description of the full training procedure: draw $k$ frequencies from $\omega$, calculate the CFs with them and perform DP sanitization, train a generator with the sanitized CFs using the minimax optimization method. The pseudo-code of the full algorithm is presented in App. E. We further give the computational complexity analysis of our algorithm in App. F.

**DP release of auxiliary information.** We also note that applying maximization (re-weighting) on a randomly selected set of frequencies would not work well. We initially fix the inverse standard deviation of the base distribution, $\omega_0$, to be the DP estimate of the mean of the pairwise distance of the data, to obtain (privately) a spectrum of frequencies that overlaps with the "good" frequencies. This is motivated by the median heuristic (Garreau et al. (2017)). Mean is estimated instead of median as its sensitivity is more tractable when considering neighboring datasets. We obtain $\Delta = 2\sqrt{d}/n$ as its sensitivity, where $d$ and $n$ are the data dimension and the number data instances respectively. See App. D for the derivation. [1] Using this DP estimate, we also investigate as a ablation study the generator performance with minimization objective only (w/o critic). The full PEARL framework includes the use of the critic to find the "best" frequencies among the selected spectrum to distinguish the distributions (and overall perform a minimax optimization).

**Evaluation Metrics.** In the main text and App. I, we show qualitative results, i.e., synthetic images (image dataset) and histograms (tabular dataset) generated with PEARL. Furthermore, for image datasets, the Fréchet Inception Distance (FID) (Heusel et al. (2017)) and Kernel Inception Distance (KID) (Bińkowski et al. (2018)) are used to evaluate the quantitative performance. For tabular data, we use the synthetic data as the training data of 10 scikit-learn classifiers (Pedregosa et al. (2011)) and evaluate the classifiers' performances on real test data. The performance indicates how well synthetic data generalize to the real data distribution and how useful synthetic data are in machine learning tasks. ROC (area under the receiver operating characteristics curve) and PRC (area under the precision recall curve) are the evaluation metrics. Definitions of FID and KID are in App. G.

### 5.2 EVALUATION DETAILS

**MNIST and Fashion-MNIST.** Privacy budget is allocated equally between the sanitization of CFs and the release of auxiliary information. [2] On a single GPU (Tesla V100-PCIE-32GB), training MNIST (with 100 epochs) requires less than 10 minutes.

---

[1] We note that DP-MERF also uses a sampling distribution. Since privacy budget is not allocated for calculating the parameters of the sampling distribution there, we fix its sampling distribution to be $\mathcal{N}(\mathbf{0}, \mathrm{diag}(\mathbf{1}))$.

[2] We expect that CF sanitization requires more privacy allocation as it is involved in doing the heavy lifting of training the model. We find that allocating around 80-90% of the budget to CF sanitization can increase the performance by around 15%. Nevertheless, we utilize an equal share of privacy budget for fairness reasons, as tuning the budget allocation would require re-using the private data and this can lead to privacy leakage.

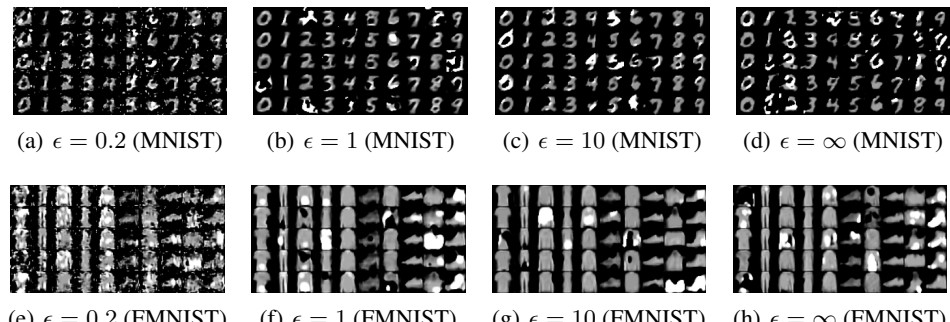

(a) $\epsilon = 0.2$ (MNIST)     (b) $\epsilon = 1$ (MNIST)     (c) $\epsilon = 10$ (MNIST)     (d) $\epsilon = \infty$ (MNIST)

(e) $\epsilon = 0.2$ (FMNIST)     (f) $\epsilon = 1$ (FMNIST)     (g) $\epsilon = 10$ (FMNIST)     (h) $\epsilon = \infty$ (FMNIST)

Figure 3: Generated MNIST and Fashion-MNIST samples for various values of $\epsilon$ and $\delta = 10^{-5}$.

| Datasets | Metrics | DPGAN | DPCGAN | DP-MERF | **Ours** (w/o critic) | **Ours** |
|---|---|---|---|---|---|---|
| MNIST | FID | $22.1 \pm 1.01$ | $17.3 \pm 2.90$ | $49.9 \pm 0.22$ | $3.79 \pm 0.06$ | $\mathbf{3.52 \pm 0.06}$ |
|  | KID ($\times 10^3$) | $573 \pm 41.2$ | $286 \pm 69.5$ | $148 \pm 46.2$ | $77.8 \pm 9.88$ | $\mathbf{70.5 \pm 10.3}$ |
| Fashion-MNIST | FID | $16.6 \pm 1.34$ | $14.61 \pm 1.75$ | $37.0 \pm 0.15$ | $1.99 \pm 0.04$ | $\mathbf{1.92 \pm 0.04}$ |
|  | KID ($\times 10^3$) | $535 \pm 61.5$ | $425 \pm 64.2$ | $1220 \pm 36.1$ | $\mathbf{24.0 \pm 6.90}$ | $26.9 \pm 6.80$ |

Table 1: FID and KID (lower is better) on image datasets at $(\epsilon, \delta) = (1, 10^{-5})$.

Some of the generated images of ours and baseline models are shown in Fig. 3, and more (including baselines') can be found in App. I. At the non-private limit, the image quality is worse than other popular non-private approaches such as GANs due to two reasons. First, projecting data to a lower dimension causes information loss. Second, our architecture does not have a discriminator-like network as in the vanilla GAN framework. However, we notice that the quality of the images does not drop much as the privacy level increases (except at $\epsilon = 0.2$, where the quality starts to degrade visibly) because the noise added to the CFs is small as it scales inversely proportional to the total sample size. It also indicates that our approach works particularly well at practical levels of privacy.

We now provide quantitative results by performing evaluation at $(\epsilon, \delta) = (1, 10^{-5})$. Note that we focus on this high privacy region despite the fact that many previous studies experimented with $\epsilon \simeq 10$, because recent analyses showed that realistic attacks can lead to privacy violations at low privacy (Jagielski et al. (2020); Nasr et al. (2021)). Particularly, the claim "values of $\epsilon$ that offer no meaningful theoretical guarantees" ($\epsilon \gg 1$) can be chosen in practice has been "refuted" in general (Nasr et al. (2021)). This motivates us to perform evaluation at $\epsilon$ with meaningful theoretical guarantees ($\epsilon \simeq 1$).

At this privacy region, we compare PEARL with models using gradient sanitization and DP-MERF (Harder et al. (2021)) as other methods do not produce usable images at single-digit $\epsilon$. [3] We run the experiment five times (with different random seeds each time), and for each time, 60k samples are generated to evaluate the FID and KID. In Table 1, the FID and KID (average and error) of DP-MERF, DPGAN, PEARL without critic, and PEARL are shown. It can be seen that PEARL outperforms DP-MERF significantly, and the critic leads to improvement in the scores.

---

[3] In the gradient sanitized generative model literature, GS-WGAN (Chen et al. (2020)) is known to be the state-of-the-art, but we are unable to train a stable model at $\epsilon = 1$. Thus, we make comparisons with other standard methods, namely DPGAN (Xie et al. (2018)) and DPCGAN (Torkzadehmahani et al. (2019)).

| | Real data | | DP-MERF | | Ours | |
|---|---|---|---|---|---|---|
| | ROC | PRC | ROC | PRC | ROC | PRC |
| LR | 0.788 | 0.681 | $0.661 \pm 0.059$ | $0.542 \pm 0.041$ | $\mathbf{0.752 \pm 0.009}$ | $\mathbf{0.641 \pm 0.015}$ |
| Gaussian NB | 0.629 | 0.511 | $0.587 \pm 0.079$ | $0.491 \pm 0.06$ | $\mathbf{0.661 \pm 0.036}$ | $\mathbf{0.537 \pm 0.028}$ |
| Bernoulli NB | 0.769 | 0.651 | $0.588 \pm 0.056$ | $0.488 \pm 0.04$ | $\mathbf{0.763 \pm 0.008}$ | $\mathbf{0.644 \pm 0.009}$ |
| Linear SVM | 0.781 | 0.671 | $0.568 \pm 0.091$ | $0.489 \pm 0.067$ | $\mathbf{0.752 \pm 0.009}$ | $\mathbf{0.640 \pm 0.015}$ |
| Decision Tree | 0.759 | 0.646 | $\mathbf{0.696 \pm 0.081}$ | $0.576 \pm 0.063$ | $0.675 \pm 0.03$ | $\mathbf{0.582 \pm 0.028}$ |
| LDA | 0.782 | 0.670 | $0.634 \pm 0.060$ | $0.541 \pm 0.048$ | $\mathbf{0.755 \pm 0.005}$ | $\mathbf{0.640 \pm 0.007}$ |
| Adaboost | 0.789 | 0.682 | $0.642 \pm 0.097$ | $0.546 \pm 0.071$ | $\mathbf{0.709 \pm 0.031}$ | $\mathbf{0.628 \pm 0.024}$ |
| Bagging | 0.772 | 0.667 | $0.659 \pm 0.06$ | $0.538 \pm 0.042$ | $\mathbf{0.687 \pm 0.041}$ | $\mathbf{0.601 \pm 0.039}$ |
| GBM | 0.800 | 0.695 | $0.706 \pm 0.069$ | $0.586 \pm 0.047$ | $\mathbf{0.709 \pm 0.029}$ | $\mathbf{0.635 \pm 0.025}$ |
| MLP | 0.776 | 0.660 | $0.667 \pm 0.088$ | $0.558 \pm 0.063$ | $\mathbf{0.744 \pm 0.012}$ | $\mathbf{0.635 \pm 0.015}$ |
| **Average** | 0.765 | 0.654 | $0.641 \pm 0.044$ | $0.536 \pm 0.034$ | $\mathbf{0.721 \pm 0.035}$ | $\mathbf{0.618 \pm 0.033}$ |

Table 2: Quantitative results for the Adult dataset evaluated at $(\epsilon, \delta) = (1, 10^{-5})$.

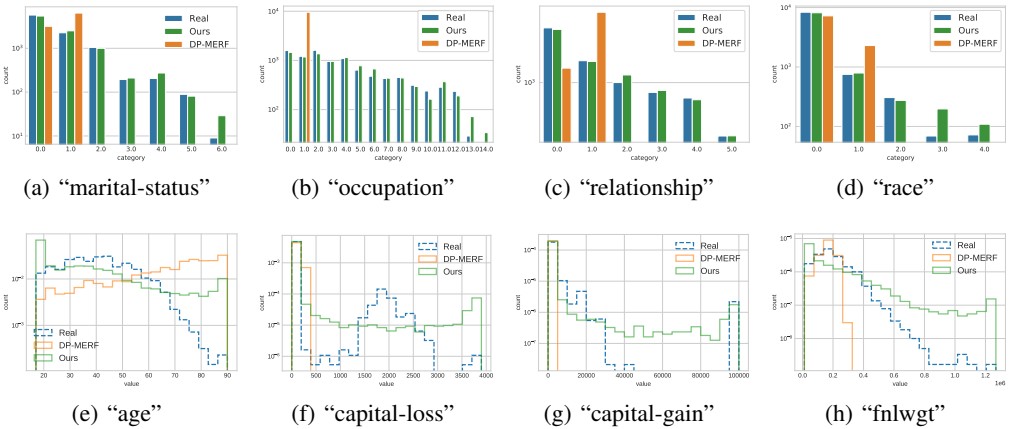

(a) "marital-status"  (b) "occupation"  (c) "relationship"  (d) "race"

(e) "age"  (f) "capital-loss"  (g) "capital-gain"  (h) "fnlwgt"

Figure 4: Histogram plots for the the Adult dataset. Evaluation is performed at $(\epsilon, \delta) = (1, 10^{-5})$.

**Adult.** The Adult dataset consists of continuous and categorical features. As data pre-processing, continuous features are scaled to $[0, 1]$. We also compare our results with DP-MERF, as DP-MERF performs best among other existing methods (e.g., DPCGAN achieves ROC and PRC around 0.5; see Table 4.) Again, auxiliary information and CFs share equal budget of privacy, and we perform evaluation at $(\epsilon, \delta) = (1, 10^{-5})$. 11k synthetic samples are generated for evaluation.

Histogram plots of the attributes comparing the real and synthetic data is shown in Fig. 4. As can be observed in the Figure, PEARL is able to model the real data better than DP-MERF, e.g., covering more modes in categorical variables with less discrepancy in the frequency of each mode. PEARL is also able to better model the descending trend of the continuous variable "age". The average ROC and PRC scores (average and error) are shown in Table 2. ROC and PRC scores based on PEARL's training data are shown to be closer to those based on real training data compared to DP-MERF. More histogram plots in higher resolution are available in App. I. Also, in App. I, we present experimental results of another tabular dataset (Credit), as well as evaluations with other metrics/tasks using the synthetic data. PEARL still performs favorably under these different conditions. Overall, we have demonstrated that PEARL is able to produce high-quality synthetic data at practical privacy levels.

## 6 CONCLUSION

We have developed a DP framework to synthesize data with deep generative models. Our approach provides synthetic samples at practical privacy levels, and sidesteps difficulties encountered in gradient sanitization methods. While we have limited ourselves to characteristic functions, it is interesting to adopt and adapt other paradigms to the PEARL framework as a future direction.

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

## A  ADDITIONAL DEFINITIONS AND PREVIOUS RESULTS

**Definition 5** (Rényi differential privacy). *A randomized mechanism $\mathcal{M}$ is said to satisfy $\varepsilon$-Rényi differential privacy of order $\lambda$, when*

$$D_\lambda(\mathcal{M}(d)\|\mathcal{M}(d')) = \frac{1}{\lambda-1}\log \mathbb{E}_{x\sim\mathcal{M}(d)}\left[\left(\frac{Pr[\mathcal{M}(d)=x]}{Pr[\mathcal{M}(d')=x]}\right)^{\lambda-1}\right] \leq \varepsilon$$

*is satisfied for any adjacent datasets $d$ and $d'$. Here, $D_\lambda(P\|Q) = \frac{1}{\lambda-1}\log\mathbb{E}_{x\sim Q}[(P(x)/Q(x))^\lambda]$ is the Rényi divergence. Furthermore, a $\varepsilon$-RDP mechanism of order $\lambda$ is also $(\varepsilon + \frac{\log 1/\delta}{\lambda-1}, \delta)$-DP.*

Next, we note that the Gaussian mechanism is $(\lambda, \frac{\lambda\Delta_2 f^2}{2\sigma^2})$-RDP (Mironov (2017)). The particular advantage of using RDP is that it gives a convenient way of tracking the privacy costs when a sequence of mechanisms is applied. More precisely, the following theorem holds (Mironov (2017)):

**Theorem 4** (RDP Composition). *For a sequence of mechanisms $M_1, ..., M_k$ s.t. $M_i$ is $(\lambda, \varepsilon_i)$-RDP $\forall i$, the composition $M_1 \circ ... \circ M_k$ is $(\lambda, \sum_i \varepsilon_i)$-RDP.*

We use the *autodp* package to keep track of the privacy budget (Wang et al. (2019)).

## B  PROOFS

### B.1  PROOF OF THM. 2

*Proof.* Let $\mathbb{P}_r$ denote the real data distribution, $\mathbb{Q}_\theta$ denote the data distribution generated by $G_\theta$ using a latent vector, $z$ sampled from a pre-determined distribution, and $\omega$ a sampling distribution. $|\cdot|$ denotes the modulus of $\cdot$. Then, the CFD between the distributions is

$$\mathcal{C}^2_\omega(\mathbb{P}_r, \mathbb{Q}_\theta) = \mathbb{E}_{\mathbf{t}\sim\omega(\mathbf{t})}\left[|\Phi_{\mathbb{P}_r}(\mathbf{t}) - \Phi_{\mathbb{Q}_\theta}(\mathbf{t})|^2\right],$$

where $\Phi_{\mathbb{P}}(\mathbf{t}) = \mathbb{E}_{\mathbf{x}\sim\mathbb{P}}\left[e^{i\mathbf{t}\cdot\mathbf{x}}\right]$ is the CF of $\mathbb{P}$. For notational brevity, we write $\Phi_{\mathbb{P}_r}(\mathbf{t})$ as $\Phi_r(\mathbf{t})$, and $\Phi_{\mathbb{Q}_\theta}(\mathbf{t})$ as $\Phi_\theta(\mathbf{t})$ in the following.

We consider the optimized CFD, $\sup_{\omega\in\Omega}\mathcal{C}^2_\omega(\mathbb{P}_r, \mathbb{Q}_\theta)$, and would like to show that it is locally Lipschitz, [4] which subsequently means that it is continuous. Since the Radamacher's theorem (Federer (2014)) implies that any locally Lipschitz function is differentiable almost everywhere, the claim of differentiability is also justified once the Lipschitz locality is proven.

We first note that the difference of two maximal functions' values is smaller or equal to the maximal difference of the two functions. Then, for any $\theta$ and $\theta'$,

$$\left|\sup_{\omega\in\Omega}\mathcal{C}^2_\omega(\mathbb{P}_r, \mathbb{Q}_\theta) - \sup_{\omega\in\Omega}\mathcal{C}^2_\omega(\mathbb{P}_r, \mathbb{Q}_{\theta'})\right| \leq \sup_{\omega\in\Omega}\left|\mathcal{C}^2_\omega(\mathbb{P}_r, \mathbb{Q}_\theta) - \mathcal{C}^2_\omega(\mathbb{P}_r, \mathbb{Q}_{\theta'})\right|. \tag{9}$$

We note that the absolute difference of any two complex values represented in terms of its real-number amplitude $(A, B)$ and phase $(\alpha, \beta)$ satisfies $|Ae^{i\alpha} - Be^{i\beta}| \leq |A| + |B|$. Writing the maximal $\omega$ as $\omega^*$, the RHS of Eq. 9 can be written as

$$\left|\mathcal{C}^2_{\omega^*}(\mathbb{P}_r, \mathbb{Q}_\theta) - \mathcal{C}^2_{\omega^*}(\mathbb{P}_r, \mathbb{Q}_{\theta'})\right| \tag{10}$$

$$= \mathbb{E}_{\mathbf{t}\sim\omega^*(\mathbf{t})}\left[(\mathcal{C}_{\omega^*}(\mathbb{P}_r, \mathbb{Q}_\theta) + \mathcal{C}_{\omega^*}(\mathbb{P}_r, \mathbb{Q}_{\theta'}))(\mathcal{C}_{\omega^*}(\mathbb{P}_r, \mathbb{Q}_\theta) - \mathcal{C}_{\omega^*}(\mathbb{P}_r, \mathbb{Q}_{\theta'}))\right]$$

$$= \mathbb{E}_{\mathbf{t}\sim\omega^*(\mathbf{t})}\left[(|\Phi_r(\mathbf{t}) - \Phi_\theta(\mathbf{t})| + |\Phi_r(\mathbf{t}) - \Phi_{\theta'}(\mathbf{t})|)(|\Phi_r(\mathbf{t}) - \Phi_\theta(\mathbf{t})| - |\Phi_r(\mathbf{t}) - \Phi_{\theta'}(\mathbf{t})|)\right]$$

$$\leq \mathbb{E}_{\mathbf{t}\sim\omega^*(\mathbf{t})}\left[(2|\Phi_r(\mathbf{t})| + |\Phi_\theta(\mathbf{t})| + |\Phi_{\theta'}(\mathbf{t})|)(|\Phi_r(\mathbf{t}) - \Phi_\theta(\mathbf{t})| - |\Phi_r(\mathbf{t}) - \Phi_{\theta'}(\mathbf{t})|)\right]$$

$$\overset{(a)}{\leq} 4\mathbb{E}_{\mathbf{t}\sim\omega^*(\mathbf{t})}\left[|\Phi_r(\mathbf{t}) - \Phi_\theta(\mathbf{t})| - |\Phi_r(\mathbf{t}) - \Phi_{\theta'}(\mathbf{t})|\right]$$

$$\overset{(b)}{\leq} 4\mathbb{E}_{\mathbf{t}\sim\omega^*(\mathbf{t})}\left[|\Phi_\theta(\mathbf{t}) - \Phi_{\theta'}(\mathbf{t})|\right],$$

---

[4] A function $f$ is locally Lipschitz if there exist constants $\delta \geq 0$ and $M \geq 0$ such that $|x - y| < \delta \rightarrow |f(x) - f(y)| \leq M \cdot |x - y|$ for all $x, y$.

where we have used $(a)$ $|\Phi_{\mathbb{P}}| \leq 1$, $(b)$ triangle inequality. By interpreting a complex number as a vector on a 2D plane, and using trigonometric arguments, one can deduce that $|e^{ia} - e^{ib}| = 2\sin(|a-b|/2) \leq |a-b|$. Then,

$$
\begin{aligned}
4\mathbb{E}_{\mathbf{t}\sim\omega^*(\mathbf{t})}\left[|\Phi_\theta(\mathbf{t}) - \Phi_{\theta'}(\mathbf{t})|\right] &= 4\mathbb{E}_{\mathbf{t}\sim\omega^*(\mathbf{t})}\left[|\mathbb{E}_z[e^{i\mathbf{t}\cdot G_\theta(z)}] - \mathbb{E}_z[e^{i\mathbf{t}\cdot G_{\theta'}(z)}]|\right] \quad (11)\\
&\overset{(c)}{\leq} 4\mathbb{E}_{\mathbf{t}\sim\omega^*(\mathbf{t})}\left[\mathbb{E}_z[|\mathbf{t}\cdot G_\theta(z) - \mathbf{t}\cdot G_{\theta'}(z)|]\right]\\
&\overset{(d)}{\leq} 4\mathbb{E}_{\mathbf{t}\sim\omega^*(\mathbf{t})}\mathbb{E}_z\left[|\mathbf{t}|\cdot|G_\theta(z) - G_{\theta'}(z)|\right].
\end{aligned}
$$

In $(c)$, we have also used the Jensen inequality. In $(d)$, the Cauchy-Schwarz inequality has been applied. As we are assuming that $G_\theta$ is a $L(\theta, z)$-Lipschitz function, we have

$$
\mathbb{E}_{\mathbf{t}\sim\omega^*(\mathbf{t})}\mathbb{E}_z\left[|\mathbf{t}|\cdot|G_\theta(z) - G_{\theta'}(z)|\right] \leq 4\mathbb{E}_{\mathbf{t}\sim\omega^*(\mathbf{t})}[|\mathbf{t}|]\cdot\mathbb{E}_z[L(\theta, z)]\cdot|\theta - \theta'|. \quad (12)
$$

Since we are also assuming that $\mathbb{E}_{\mathbf{t}\sim\omega^*(\mathbf{t})}[|\mathbf{t}|], \mathbb{E}_z[L(\theta, z)] \leq \infty$, we have shown that $\sup_{\omega\in\Omega}\mathcal{C}_\omega$ is locally Lipschitz, as required. It is therefore continuous and differentiable almost everywhere, as discussed above. □

## B.2 PROOF OF THM. 3

*Proof.* We denote $\mathbf{x}_n \sim \mathbb{P}_n$ and $\mathbf{x} \sim \mathbb{P}$, and $\omega^*$ the maximal function of $\omega$. Notice that

$$
\begin{aligned}
\mathcal{C}^2_{\omega^*}(\mathbb{P}_n, \mathbb{P}) &= \mathbb{E}_{\mathbf{t}\sim\omega^*(\mathbf{t})}\left[\left|\mathbb{E}_{\mathbf{x}_n}\left[e^{i\mathbf{t}\cdot\mathbf{x}_n}\right] - \mathbb{E}_{\mathbf{x}}\left[e^{i\mathbf{t}\cdot\mathbf{x}}\right]\right|^2\right]\\
&= \mathbb{E}_{\mathbf{t}\sim\omega^*(\mathbf{t})}\left[\left|\mathbb{E}_{\mathbf{x}_n}\left[e^{i\mathbf{t}\cdot\mathbf{x}_n}\right] - \mathbb{E}_{\mathbf{x}}\left[e^{i\mathbf{t}\cdot\mathbf{x}}\right]\right|\cdot\left|\mathbb{E}_{\mathbf{x}_n}\left[e^{i\mathbf{t}\cdot\mathbf{x}_n}\right] - \mathbb{E}_{\mathbf{x}}\left[e^{i\mathbf{t}\cdot\mathbf{x}}\right]\right|\right]\\
&\overset{(a)}{\leq} 2\,\mathbb{E}_{\mathbf{t}\sim\omega^*(\mathbf{t})}\left[\left|\mathbb{E}_{\mathbf{x}_n}\left[e^{i\mathbf{t}\cdot\mathbf{x}_n}\right] - \mathbb{E}_{\mathbf{x}}\left[e^{i\mathbf{t}\cdot\mathbf{x}}\right]\right|\right]\\
&\overset{(b)}{\leq} 2\,\mathbb{E}_{\mathbf{t}\sim\omega^*(\mathbf{t})}\,\mathbb{E}_{\mathbf{x}_n,\mathbf{x}}\left[\left|e^{i\mathbf{t}\cdot\mathbf{x}_n} - e^{i\mathbf{t}\cdot\mathbf{x}}\right|\right]\\
&\overset{(c)}{\leq} 2\,\mathbb{E}_{\mathbf{t}\sim\omega^*(\mathbf{t})}\left[|\mathbf{t}|\right]\mathbb{E}_{\mathbf{x}_n,\mathbf{x}}\left[|\mathbf{x}_n - \mathbf{x}|\right]. \quad (13)
\end{aligned}
$$

Here, $(a)$ uses $|Ae^{i\alpha} - Be^{i\beta}| \leq |A| + |B|$ as argued above Eq. 10; $(b)$ uses Jensen inequality; $(c)$ uses the argument $|e^{ia} - e^{ib}| = 2\sin(|a-b|/2) \leq |a-b|$, given above Eq. 11. Then, by weak convergence equivalence (Klenke (2013)), the RHS of Eq. 13 approaches zero as $\mathbb{P}_n \overset{D}{\longrightarrow} \mathbb{P}$, hence proving the theorem. □

## B.3 PROOF OF LEMMA. 1

*Proof.* Recall that for any two distributions, $\omega(\mathbf{t})$, $\omega_0(\mathbf{t})$ and any function $f(\mathbf{t})$,

$$
\begin{aligned}
\mathbb{E}_\omega[f(\mathbf{t})] &= \int f(\mathbf{t})\omega(\mathbf{t})d\mathbf{t}\\
&= \int f(\mathbf{t})\frac{\omega(\mathbf{t})}{\omega_0(\mathbf{t})}\omega_0(\mathbf{t})d\mathbf{t}\\
&= \mathbb{E}_{\omega_0}[f(\mathbf{t})\frac{\omega(\mathbf{t})}{\omega_0(\mathbf{t})}].
\end{aligned}
$$

Hence, $\mathbb{E}_{\omega(\mathbf{t})}[f(\mathbf{t})] = \mathbb{E}_{\omega_0}[f(\mathbf{t})\frac{\omega(\mathbf{t})}{\omega_0(\mathbf{t})}]$. Let $\omega^*$ be the maximal probability distribution. It is then clear that $\widehat{\mathbb{E}}_\omega[f(\mathbf{t})] \to \mathbb{E}_\omega[f(\mathbf{t})]$ implies $\widehat{\mathbb{E}}_{\omega_0}[f(\mathbf{t})\frac{\omega^*(\mathbf{t})}{\omega_0(\mathbf{t})}] \to \mathbb{E}_{\omega^*}[f(\mathbf{t})]$, as desired. □

## C EXPERIMENTAL SETUP OF TWO-SAMPLE TESTING ON SYNTHETIC DATA

Let us describe in more detail the experiment presented in Sec. 4. Data are generated from two unit-variance multivariate Gaussian distributions $\mathbb{P}, \mathbb{Q}$, where all dimensions but one have zero mean ($\mathbb{P} \sim \mathcal{N}(\mathbf{0}_d, \mathbf{I}_d)$, $\mathbb{Q} \sim \mathcal{N}((1, 0, \ldots, 0)^\top, \mathbf{I}_d)$). We wish to conduct a two-sample test using the CF to

distinguish between the two distributions, which gets more difficult as the dimensionality increases. We test if the null hypothesis where the samples are drawn from the same distribution is rejected.

Three sets of frequencies are considered. The number of frequncies in each set is set to 20. The first set is an "unoptimized" set of frequencies. The first dimension of all but one frequency has the value of zero. Other dimensions have values generated randomly from a zero-mean unit-variance multivariate Gaussian distribution. We denote the frequency with non-zero value in the first dimension by $\mathbf{t}_0$ without loss of generality. A "normal" set of frequencies is also considered for comparison, where the frequencies of all dimensions are sampled randomly from a multivariate Gaussian distributions. Finally, we consider an "optimized" set of frequencies, where from the "unoptimized" set of frequencies, only $\mathbf{t}_0$ is selected to be used for two-sample testing. In other words, we re-weight the set of frequencies such that all but $\mathbf{t}_0$ has zero weight. 1,000 samples are generated from each of $\mathbb{P}$ and $\mathbb{Q}$. We repeat the problem for 100 trials to obtain the rejection rate (and repeat the whole experiment 5 times to get the error bar).

## D  DP ESTIMATE OF THE MEAN OF PAIRWISE DISTANCE

Median heuristic is applied widely in kernel methods applications to determine the bandwidth of the radial basis function (RBF) kernels (Garreau et al. (2017)). The bandwidth is taken to be the median of all pairwise distances of data samples. Here we give a DP estimation of the mean instead as the calculation of mean is more tractable. Let $\mathbf{x}$ be samples of a certain data distribution of dimension $d$ and assume that the values lie in $[0, 1]^d$. Given $n$ samples, there is a total of $n(n-1)/2$ pairwise distance pairs. Then, the mean of the pairwise distance of samples is

$$\overline{D}_n(\mathbf{x}) = \frac{2}{n(n-1)} \sum_{i \neq j}^{n} \|\mathbf{x}_i - \mathbf{x}_j\|_2.$$

where $\|\cdot\|_2$ indicates the Euclidean norm.

Consider a pair of neighboring datasets, $\mathcal{D}, \mathcal{D}'$. Without loss of generality, let $\mathbf{x}_n \neq \mathbf{x}'_n$ and $\mathbf{x}_i = \mathbf{x}'_i$ for $i \neq n$. Then, the sensitivity of $\overline{D}_n(\mathbf{x})$ is

$$\Delta_{\overline{D}_n(\mathbf{x})} = \max_{\mathcal{D}, \mathcal{D}'} \left\| \frac{2}{n(n-1)} \sum_{i \neq j}^{n} \|\mathbf{x}_i - \mathbf{x}_j\|_2 - \frac{2}{n(n-1)} \sum_{i \neq j}^{n} \|\mathbf{x}'_i - \mathbf{x}'_j\|_2 \right\|_2$$

$$\overset{(a)}{=} \frac{2}{n(n-1)} \max_{\mathcal{D}, \mathcal{D}'} \left\| \sum_{i=1}^{n-1} \|\mathbf{x}_i - \mathbf{x}_n\|_2 - \sum_{i=1}^{n-1} \|\mathbf{x}_i - \mathbf{x}'_n\|_2 \right\|_2$$

$$\overset{(b)}{=} \frac{2}{n(n-1)} \cdot (n-1)\sqrt{d}$$

$$= \frac{2\sqrt{d}}{n}.$$

In (a), we cancel out all terms unrelated to $\mathbf{x}_n$ or $\mathbf{x}'_n$. In (b), we use the fact that $\|\mathbf{x}_i - \mathbf{x}_n\|_2$ and $\|\mathbf{x}_i - \mathbf{x}'_n\|_2$ lie in $[0, \sqrt{d}]$. After obtaining the sensitivity, one can then applies the Gaussian mechanism as in Eq. 6 to obtain the DP estimate of the mean of the pairwise distance of samples.

## E  TRAINING ALGORITHM

The pseudo-code of the proposed training algorithm is given in Algorithm 1.

## F  COMPUTATIONAL COMPLEXITY ANALYSIS

PEARL is trained based on deep learning methods, which is efficient computation-wise using a GPU. Nevertheless, let us give a computational complexity analysis in a traditional sense by ignoring (partially) the parallel computing capability.

---

**Algorithm 1:** PEARL Training

---

**Input:** Sensitive data $\{\mathbf{x}\}_{i=1}^n$, differential privacy noise scale $\sigma_{\mathrm{DP}}$, number of frequencies $k$, base sampling distribution variance $\boldsymbol{\sigma}_0$, training iterations $T$, learning rates $\eta_C$ and $\eta_G$, number of generator iterations per critic iteration $n_{gen}$, batch size $B$, latent distribution $P_z$

**Output:** Differentially private generator $G_\theta$

1   Obtain auxiliary information (e.g., base sampling distribution variance $\boldsymbol{\sigma}_0$);

2   Sample frequencies $\{\mathbf{t}\}_{i=1}^k$ with $\mathbf{t} \sim \mathcal{N}(\mathbf{0}, \mathrm{diag}(\boldsymbol{\sigma}_0))$;

3   **for** $i$ **in** $\{1, ..., k\}$ **do**

4     $\widehat{\Phi}_{\mathbb{P}}(\mathbf{t}_i) = \frac{1}{n}\sum_{j=1}^n e^{i\mathbf{t}_i \cdot \mathbf{x}_j}$

5     $\widetilde{\Phi}_{\mathbb{P}}(\mathbf{t}_i) = \widehat{\Phi}_{\mathbb{P}}(\mathbf{t}_i) + \mathcal{N}(0, \Delta_{\widehat{\phi}(\mathbf{x})}^2 \sigma_{\mathrm{DP}}^2 I)$

6   **end**

7   Accumulate privacy cost $\epsilon$;

8   $\widetilde{\phi}(\mathbf{x}) \leftarrow (\widetilde{\Phi}_{\mathbb{P}}(\mathbf{t}_1), \ldots, \widetilde{\Phi}_{\mathbb{P}}(\mathbf{t}_k))^\top$

9   Initialize generator $G_\theta$, sampling distribution variance $\boldsymbol{\sigma}$ ;

10   **for** $step$ **in** $\{1, ..., T\}$ **do**

11     **for** $t$ **in** $\{1, ..., n_{gen}\}$ **do**

12       Sample batch $\{\mathbf{z}_i\}_{i=1}^B$ with $\mathbf{z}_i \sim P_z$;

13       **for** $i$ **in** $\{1, ..., k\}$ **do**

14         $\widehat{\Phi}_{\mathbb{Q}}(\mathbf{t}_i) = \frac{1}{B}\sum_{j=1}^B e^{i\mathbf{t}_i \cdot G_\theta(\mathbf{z}_j)}$

15       **end**

16       $\widehat{\phi}(\mathbf{z}) \leftarrow (\widehat{\Phi}_{\mathbb{Q}}(\mathbf{t}_1), \ldots, \widehat{\Phi}_{\mathbb{Q}}(\mathbf{t}_k))^\top$

17       $\theta \leftarrow \theta - \eta_G \cdot \nabla_\theta \widehat{\mathcal{C}}_\omega^2(\widetilde{\phi}(\mathbf{x}), \widehat{\phi}(\mathbf{z}))$

18     **end**

19     Sample batch $\{\mathbf{z}_i\}_{i=1}^B$ with $\mathbf{z}_i \sim P_z$;

20     **for** $i$ **in** $\{1, ..., k\}$ **do**

21       $\widehat{\Phi}_{\mathbb{Q}}(\mathbf{t}_i) = \frac{1}{B}\sum_{j=1}^B e^{i\mathbf{t}_i \cdot G_\theta(\mathbf{z}_j)}$

22     **end**

23     $\widehat{\phi}(\mathbf{z}) \leftarrow (\widehat{\Phi}_{\mathbb{Q}}(\mathbf{t}_1), \ldots, \widehat{\Phi}_{\mathbb{Q}}(\mathbf{t}_k))^\top$

24     $\boldsymbol{\sigma} \leftarrow \boldsymbol{\sigma} + \eta_C \cdot \nabla_{\boldsymbol{\sigma}} \widehat{\mathcal{C}}_\omega^2(\widetilde{\phi}(\mathbf{x}), \widehat{\phi}(\mathbf{z}))$

25   **end**

26   **Return** $G_\theta$

---

**Time complexity.** We sample $k$ frequencies and calculate its inner product with data points (of dimension $d$), make a summation with respect to $n$ data points to build $k$ CFs. Since inner product takes $d$ steps, summation $n$ steps, and we do it for $k$ times, the time complexity of generating CFs is $nkd$. We additionally train the generative model, requiring $t_{tr}$. Note that unlike graphical models like PrivBayes, which gradually add nodes to the Bayesian network, our method can be parallelized efficiently with GPU. As we sample $n$ records from the generative model during data generation, the time complexity of dataset generation is $n$ times the inference time, $t_{inf}$. The total training and inference time is $O(nkd + nt_{inf} + t_{tr})$.

**Space complexity.** We need to store the k CF values during training. We also need to store the generative model with $m$ parameters. The space complexity is $O(k + m)$.

## G   EVALUATION METRICS

We utilize evaluation metrics commonly used to evaluate GAN's performance, namely Fréchet Inception Distance (FID) (Heusel et al. (2017)) and Kernel Inception Distance (KID) (Bińkowski et al. (2018)). FID corresponds to computing the Fréchet distance between the Gaussian fits of the Inception features obtained from real and fake distributions. KID is the calculation of the MMD of the Inception features between real and fake distributions using a polynomial kernel of degree 3.

The precision definitions are as follows. Let $\{\mathbf{x}^r_i\}_{i=1}^n$ be samples from the real data distribution $\mathbb{P}_r$ and $\{\mathbf{x}^\theta_i\}_{i=1}^m$ be samples from the generated data distribution $\mathbb{Q}_\theta$. The corresponding feature vectors extracted from a pre-trained network (LeNet in our case) are $\{\mathbf{z}^r_i\}_{i=1}^n$ and $\{\mathbf{z}^\theta_i\}_{i=1}^m$ respectively. The FID and KID are defined as

$$\text{FID}(\mathbb{P}_r, \mathbb{Q}_\theta) = \|\mu_r - \mu_\theta\|_2^2 + \text{Tr}(\Sigma_r + \Sigma_\theta - 2(\Sigma_r \Sigma_\theta)^{1/2}), \tag{14}$$

$$\begin{aligned}
\text{KID}(\mathbb{P}_r, \mathbb{Q}_\theta) = & \frac{1}{n(n-1)} \sum_{i=1}^n \sum_{j=1, j \neq i}^n \left[ \kappa(\mathbf{z}^r_i, \mathbf{z}^r_j) \right] \\
& + \frac{1}{m(m-1)} \sum_{i=1}^m \sum_{j=1, j \neq i}^m \left[ \kappa(\mathbf{z}^\theta_i, \mathbf{z}^\theta_j) \right] \\
& - \frac{2}{mn} \sum_{i=1}^n \sum_{j=1}^m \left[ \kappa(\mathbf{z}^r_i, \mathbf{z}^\theta_j) \right],
\end{aligned} \tag{15}$$

where $(\mu_r, \Sigma_r)$ and $(\mu_\theta, \Sigma_\theta)$ are the sample mean & covariance matrix of the inception features of the real and generated data distributions, and $\kappa$ is a polynomial kernel of degree 3:

$$\kappa(\mathbf{x}, \mathbf{y}) = \left( \frac{1}{d} \mathbf{x} \cdot \mathbf{y} + 1 \right)^3, \tag{16}$$

where $d$ is the dimensionality of the feature vectors. We compute FID with 10 bootstrap resamplings and KID by sampling 100 elements without replacement from the whole generated dataset.

## H  IMPLEMENTATION DETAILS

**Datasets.** For MNIST and Fashion-MNIST, we use the default train subset of the `torchvision`[5] library for training the generator, and the default subset for evaluation. For Adult, we follow the preprocessing procedure in Harder et al. (2021) to make the dataset more balanced by downsampling the class with the most number of samples.

**Neural networks.** The generator for image datasets has the following network architecture:

- fc → bn → fc → bn → upsamp → relu → upconv → sigmoid,

where fc, bn, upsamp, relu, upconv, sigmoid refers to fully connected, batch normalization, 2D bilinear upsampling, ReLU, up-convolution, and Sigmoid layers respectively.

For tabular dataset, we use the following architecture:

- fc → bn → relu → fc → bn → relu → fc → tanh/softmax,

where tanh and softmax are the Tanh and softmax layers respectively. Network output corresponding to the continuous attribute is passed through the Tanh layer, whereas network output corresponding to the categorical attribute is passed through the softmax layer, where the category with the highest value from the softmax output is set to be the generated value of the categorical attribute. We train both networks conditioned on the class labels. For DP-MERF, we use the same architectures for fair comparisons.

**Hyperparameters.** We use Adam optimizer with learning rates of 0.01 for both the minimization and maximization objectives. Batch size is 100 (1,100) for the image datasets (tabular dataset). The number of frequencies is set to 1,000 (3,000) for MNIST and tabular datasets (Fashion-MNIST). The training iterations are 6,000, 3,000, and 8,000 for MNIST, Fashion-MNIST, and tabular datasets respectively.

---

[5]https://pytorch.org/vision/stable/index.html

| Datasets | Methods | ROC | PRC | Range | Marginal | MMD |
|---|---|---|---|---|---|---|
| Adult | DP-MERF | 0.64 | 0.54 | 0.10 | 1.37 | $2 \times 10^{-3}$ |
| | **Ours** | **0.72** | **0.62** | **0.06** | **0.71** | $\mathbf{1 \times 10^{-7}}$ |
| Credit | DP-MERF | 0.66 | 0.26 | 0.049 | 0.95 | $2 \times 10^{-5}$ |
| | **Ours** | **0.84** | **0.68** | **0.026** | **0.72** | $\mathbf{1 \times 10^{-8}}$ |

Table 3: Evaluation with ROC, PRC (higher is better), range query, marginal release and MMD (lower is better) on tabular datasets at $(\epsilon, \delta) = (1, 10^{-5})$.

| | Real data | | DPCGAN | | Ours | |
|---|---|---|---|---|---|---|
| | ROC | PRC | ROC | PRC | ROC | PRC |
| LR | 0.788 | 0.681 | $0.501 \pm 0.001$ | $0.484 \pm 0.001$ | $\mathbf{0.752 \pm 0.009}$ | $\mathbf{0.641 \pm 0.015}$ |
| Gaussian NB | 0.629 | 0.511 | $0.499 \pm 0.005$ | $0.483 \pm 0.003$ | $\mathbf{0.661 \pm 0.036}$ | $\mathbf{0.537 \pm 0.028}$ |
| Bernoulli NB | 0.769 | 0.651 | $0.482 \pm 0.159$ | $0.498 \pm 0.091$ | $\mathbf{0.763 \pm 0.008}$ | $\mathbf{0.644 \pm 0.009}$ |
| Linear SVM | 0.781 | 0.671 | $0.501 \pm 0.001$ | $0.484 \pm 0.001$ | $\mathbf{0.752 \pm 0.009}$ | $\mathbf{0.640 \pm 0.015}$ |
| Decision Tree | 0.759 | 0.646 | $0.599 \pm 0.078$ | $0.546 \pm 0.056$ | $\mathbf{0.675 \pm 0.03}$ | $\mathbf{0.582 \pm 0.028}$ |
| LDA | 0.782 | 0.670 | $0.501 \pm 0.002$ | $0.484 \pm 0.002$ | $\mathbf{0.755 \pm 0.005}$ | $\mathbf{0.640 \pm 0.007}$ |
| Adaboost | 0.789 | 0.682 | $0.523 \pm 0.065$ | $0.5 \pm 0.036$ | $\mathbf{0.709 \pm 0.031}$ | $\mathbf{0.628 \pm 0.024}$ |
| Bagging | 0.772 | 0.667 | $0.546 \pm 0.082$ | $0.516 \pm 0.058$ | $\mathbf{0.687 \pm 0.041}$ | $\mathbf{0.601 \pm 0.039}$ |
| GBM | 0.800 | 0.695 | $0.544 \pm 0.101$ | $0.518 \pm 0.066$ | $\mathbf{0.709 \pm 0.029}$ | $\mathbf{0.635 \pm 0.025}$ |
| MLP | 0.776 | 0.660 | $0.503 \pm 0.004$ | $0.486 \pm 0.004$ | $\mathbf{0.744 \pm 0.012}$ | $\mathbf{0.635 \pm 0.015}$ |

Table 4: Quantitative results for the Adult dataset evaluated at $(\epsilon, \delta) = (1, 10^{-5})$.

# I ADDITIONAL RESULTS

## I.1 ADDITIONAL TASKS AND DATASET FOR TABULAR DATA

We here present more results relevant to PEARL's generative modeling of tabular data. We first compare PEARL with DPCGAN in Table 4 using the same classifiers as in the main text.

Three additional metrics/tasks are utilized: **Range query**. 1000 range queries containing three attributes are sampled randomly. The average $l_1$ error between the original and synthetic datasets is evaluated. **Marginal release**. All 2-way marginals are calculated and the average $l_1$ error is evaluated. **MMD**. The max mean discrepancy (MMD) between the original and synthetic dataset is evaluated. We use the MMD because it is a powerful measure that could in principle detect any discrepancy between two distributions. A Gaussian kernel is used in the evaluation.

Additionally, we perform evaluation on another tabular dataset: Credit (Dal Pozzolo et al. (2015)). Our results comparing with DP-MERF as shown in Table 3 demonstrate that PEARL is competitive with various tasks and datasets.

## I.2 MNIST AND FASHION-MNIST IMAGES

Fig. 5 and Fig. 6 show more enlarged images of MNIST and Fashion-MNIST at various level of $\epsilon$ generated with PEARL. We also show images generated by PEARL but without critic, and DPGAN. Note that we are unable to generate meaningful images from DP-MERF after taking privacy into account appropriately (see Sec. 5).

## I.3 ADULT HISTOGRAMS

We show more histogram plots for continuous and categorical attributes comparing our method with DP-MERF in Fig. 9 and Fig. 10.

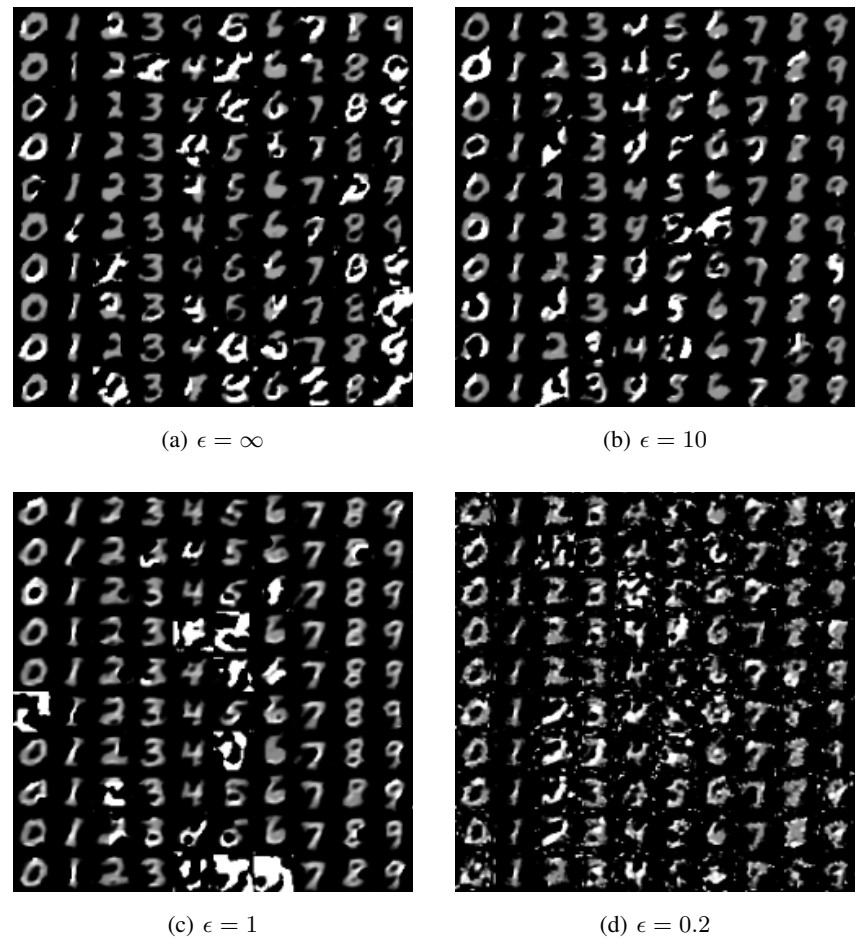

(a) $\epsilon = \infty$

(b) $\epsilon = 10$

(c) $\epsilon = 1$

(d) $\epsilon = 0.2$

Figure 5: Additional generated MNIST images at various $\epsilon$.

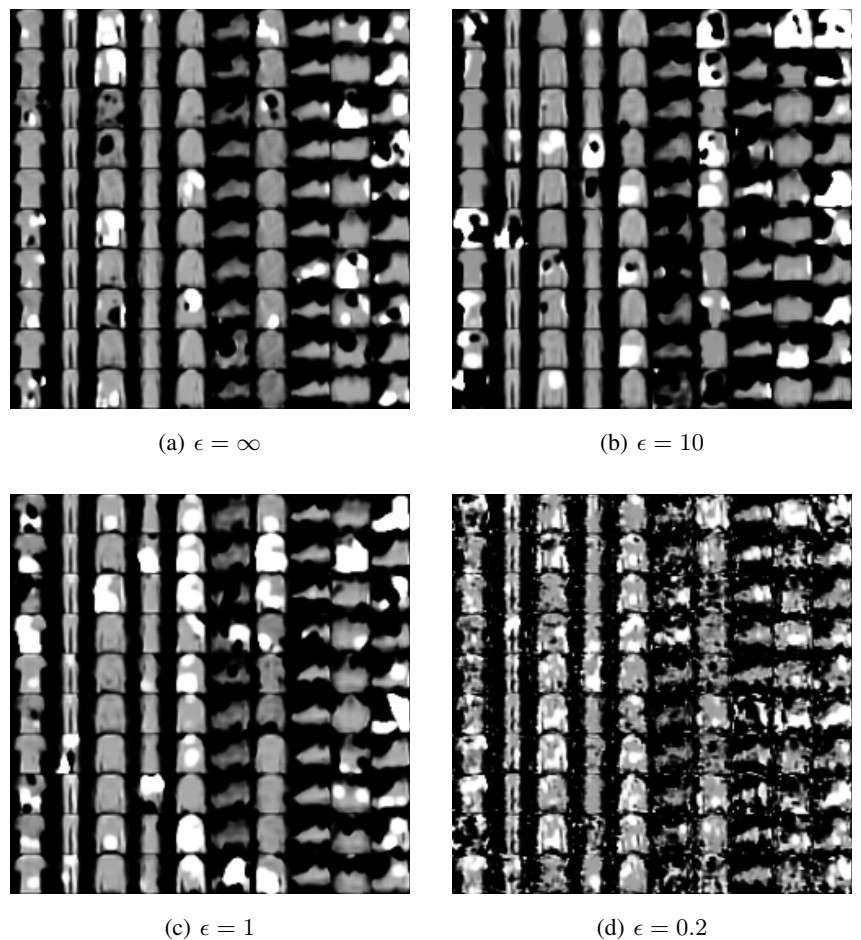

(a) $\epsilon = \infty$                        (b) $\epsilon = 10$

(c) $\epsilon = 1$                        (d) $\epsilon = 0.2$

Figure 6: Additional generated Fashion-MNIST images at various $\epsilon$.

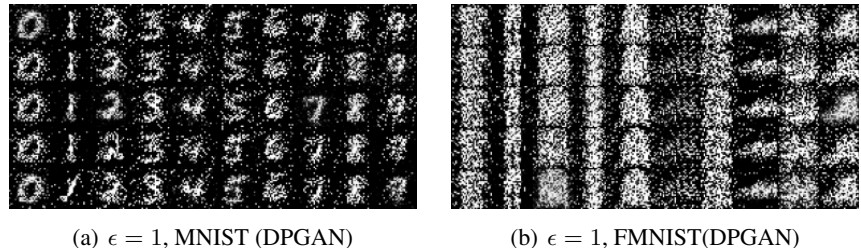

(a) $\epsilon = 1$, MNIST (DPGAN)           (b) $\epsilon = 1$, FMNIST(DPGAN)

Figure 7: Images generated by DPGAN at $\epsilon = 1$.

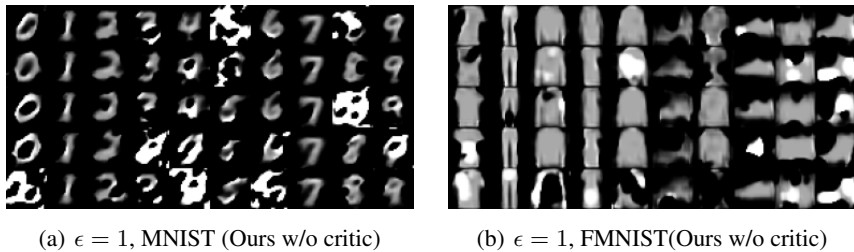

(a) $\epsilon = 1$, MNIST (Ours w/o critic)      (b) $\epsilon = 1$, FMNIST(Ours w/o critic)

Figure 8: Images generated by our proposal (PEARL) but without critic at $\epsilon = 1$.

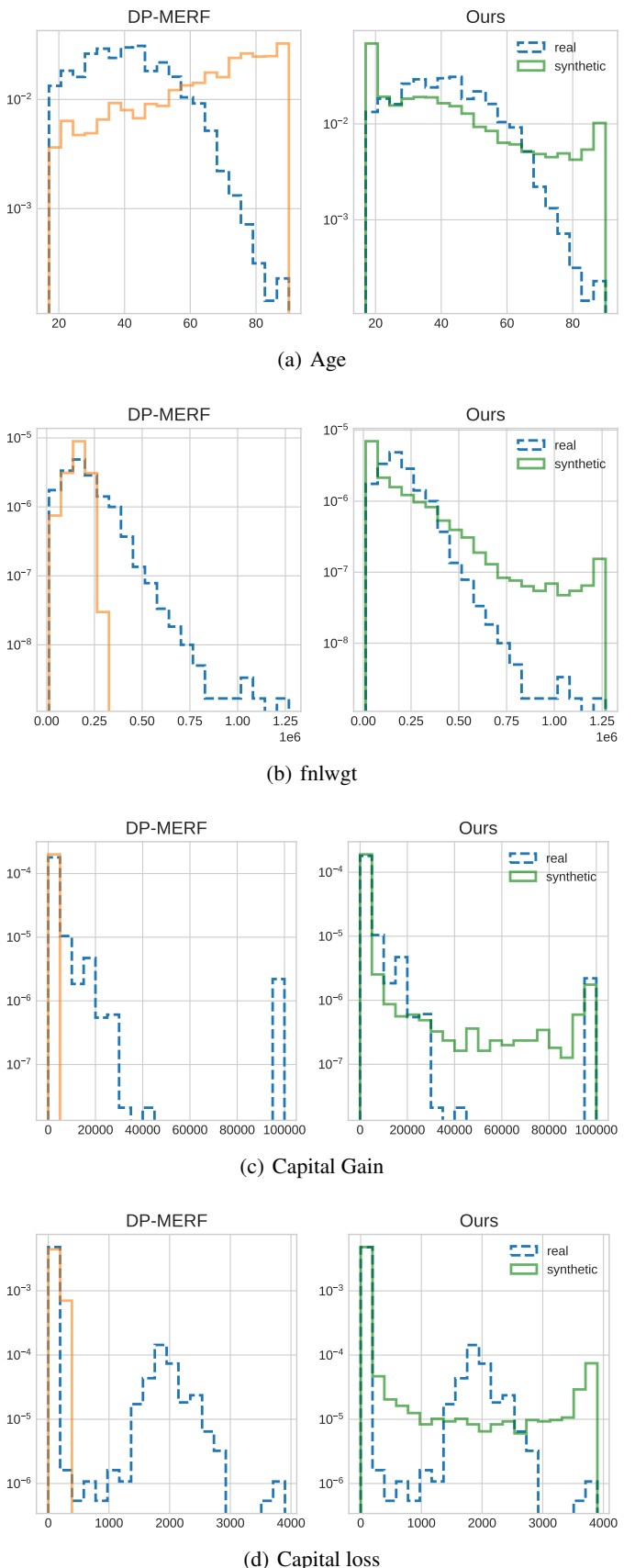

Figure 9: Histograms of various continuous attributes of Adult dataset comparing real and synthetic data.

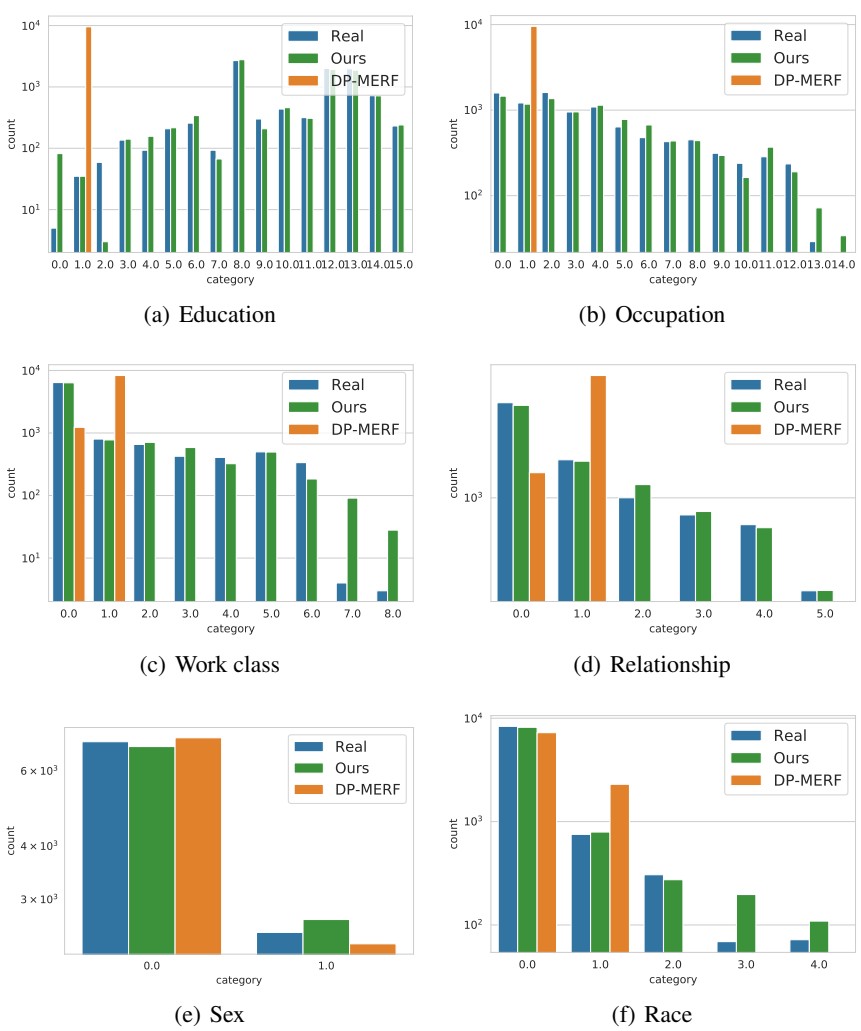

Figure 10: Histograms of various categorical attributes of Adult dataset comparing real and synthetic data.

