# OpenReview forum: "PEARL: Data Synthesis via Private Embeddings and Adversarial Reconstruction Learning"
_ICLR.cc/2022/Conference — ICLR 2022 Poster_

### Official Review · Reviewer_Xui7 · 2021-11-02

**Correctness:** 3
**Technical Novelty And Significance:** 3
**Empirical Novelty And Significance:** 3
**Recommendation:** 6
**Confidence:** 3

**Main Review:**

The strengths and weaknesses of this paper are summarized as follows:
Strengths:
+ The problem studied in this paper is important and needs to be solved in data synthesis
+ Good writing
+ Sufficient theoretical analysis

Weaknesses:
- Need to include more related work that is highly important
- Need more justifications about the novelty claims
- Need to add more benchmark datasets
- Need to add more advanced baselines
- Lack of complexity analysis

Comments:
1. The following important references are missing:

[1] Zhang Z, Wang T, Li N, et al. Privsyn: Differentially private data synthesis[C]//30th {USENIX} Security Symposium ({USENIX} Security 21). 2021.

[2] He, Xi, et al. "DPT: differentially private trajectory synthesis using hierarchical reference systems." Proceedings of the VLDB Endowment 8.11 (2015): 1154-1165.

[3] Cai, Kuntai, et al. "Data synthesis via differentially private Markov random fields." Proceedings of the VLDB Endowment 14.11 (2021): 2190-2202.

2. How to implement privacy-preserving optimization in this paper is vague. It would be better if the authors could add more details, especially how to perform one-shot sampling. In addition, the authors need to explain how the one-shot sampling strategy guarantees privacy.
3. It would be better if the authors could add 1-2 more public datasets to verify the performance of the proposed framework. For example, the Colorado dataset is the census dataset of Colorado State in 1940, which is used in the final round of the NIST challenge.
4. It would be better if the authors could add 1-2 more baseline schemes (e.g., PrivBayes, PGM, and PrivSyn) to verify the performance of the proposed framework.
5. Like [1], it gives a detailed analysis of the computational complexity of the algorithm. Therefore, it would be better if the authors could give an analysis of the computational complexity of the proposed algorithm.
6. In [1], the authors evaluated the statistical performance of the synthesized datasets on three data analysis tasks (i.e., Marginal Release, Range Query, and Classification). Therefore, it would be better if the authors could add more tasks to verify the proposed algorithm.
7. The editorial quality of this paper is not always satisfactory. It contains quite a lot of inconsistent/non-precise descriptions, as also reflected in the above comments.


**Summary Of The Paper:**

This paper studied an important topic in the field of data synthesis: how to train a private deep generative model without reusing the original data. In this paper, the authors proposed a new framework that uses deep generative models to synthesize data in different private ways. Unlike popular gradient cleaning methods, the framework proposed in this paper does not incur additional privacy costs or model constraints. In addition, in order to avoid the problem of reduced privacy guarantee as training iterations increase, this paper used feature functions and adversarial re-weighting objectives to solve the above problems. Both theory and extensive case studies verify the performance of the proposed framework.

**Summary Of The Review:**

The paper tackles a very interesting problem, and the many technical considerations and the experiments on various datasets and comparisons with existing data synthesis techniques are commendable. Some issues (unclear privacy-preserving optimization method, need to add baselines and benchmark datasets) still prevent me from recommending complete acceptance. More clarifications are necessary, and by adding some more "realistic" experiments, I believe the paper could be turned into a significant submission of ICLR. I recommend a "5", but my score can be easily increased to 6 by addressing the many clarifications expressed in my review. Further experiments and the concrete use-case would further improve my score.

---

> ### Author Response · Authors · 2021-11-22
> **Response to Reviewer**
>
> We thank the Reviewer for helpful comments. Please find the following our replies to the comments.
>
> 1) The following important references are missing ...
>
> We thank the reviewer for the suggestions and we have added the references correspondingly.
>
> 2) How to implement privacy-preserving optimization in this paper is vague. It would be better if the authors could add more details, especially how to perform one-shot sampling. In addition, the authors need to explain how the one-shot sampling strategy guarantees privacy.
>
> We apologize for the lack of clarity.
> Let us explain below.
>
> We first draw k frequencies of d dimension (assuming that data is of dimension d) from a distribution (e.g., Gaussian distribution),t_i
> We use the data to calculate the characteristic functions using the frequencies for just once and add noises to them. This is what we meant by one-shot sampling, which is, in contrast to DP-SGD, which requires multiple (batched) data accesses for multi-epoch or iteration training. Here, differential privacy is guaranteed by the addition of noises.
> We further introduce an adversarial optimization objective (see Eq. 1) which acts on the post-processed characteristic functions to enhance the performance of generator.
> By post-processing properties, this adversarial optimization does not further degrade the privacy budget, and the whole training process is therefore "one-shot" (only requires single access to data).
>
> Please let us know if there is anything unclear about the optimization procedure (We have also make modification in text, including a high-level description of the algorithm in Section 5.1, to further clarify our procedure).
>
> 3) and 4.
> It would be better if the authors could add 1-2 more public datasets / 1-2 more baseline schemes (e.g., PrivBayes..
>
> During rebuttal, we have added another dataset, Credit, and another baseline, DPCGAN. Some additional results are given in Appendix I.
> We did not evaluate graphical model methods such as PrivBayes because they cannot deal with continuous attributes naturally (e.g., image data or tabular data with continuous attributes) like generative models including PEARL (which is able to model both continuous and discrete attributes simultaneously). We mentioned this in the related works part of Introduction.
>
> 5. Like [1], it gives a detailed analysis of the computational complexity of the algorithm...
>
> We would like to emphasize that our method is based on deep learning, which is efficient computation-wise using a GPU (the overall process takes less than 10 minutes as mentioned in the text). Nevertheless in the following, let us give a computational complexity analysis in a traditional sense by ignoring (partially) the parallel computing capability.
>
> Time complexity:
>
>  We sample $k$ frequencies and calculate its inner product with data points (of dimension $d$), make a summation with respect to $n$ data points to build $k$ CFs.
> Since inner product takes $d$ steps, summation $n$ steps, and we do it for $k$ times, the time complexity of generating CFs is $nkd$.
> We additionally train the generative model, requiring $t_{tr}$.
> Note that unlike graphical models like PrivBayes, which gradually add nodes to the Bayesian network, our method can be parallelized efficiently with GPU.
> As we sample $n$ records from the generative model during data generation, the  time complexity of dataset generation is $n$ times the inference time, $t_{inf}$.
> The total training and inference time is $O(nkd + nt_{inf} + t_{tr})$.
>
> Space complexity:
>
> We need to store the k CF values during training. We also need to store the generative model with $m$ parameters. The space complexity is $O(k+m)$.
>
> We added the above arguments in Appendix F.
>
> 6. In [1], the authors evaluated the statistical performance of the synthesized datasets on three data analysis tasks (i.e., Marginal Release...
>
> We have added additional evaluation tasks: marginal release, range query and MMD.
> We use MMD because compared to marginal release and range query which only measure partial discrepancy (e.g., only 2 attributes are measured at one time when comparing the 2-way marginals), it is a powerful measure that can in principle detect the distributional discrepancy between two datasets.
> We show in Appendix I that PEARL is still competitive.
> Let us also clarify why we perform classification tasks in our paper. We trained machine learning models with synthetic data to evaluate their performance against real test data. The performance indicates how well synthetic data generalize to the real data distribution and how useful synthetic data are in machine learning tasks.

---

### Official Review · Reviewer_kXZv · 2021-11-02

**Correctness:** 3
**Technical Novelty And Significance:** 3
**Empirical Novelty And Significance:** 3
**Recommendation:** 6
**Confidence:** 3

**Main Review:**

PEARL, a method to generate synthetic data with privacy preservation is proposed. Please see my detailed comments below:

1. I am a bit confused by the auxiliary information release, is the total privacy budget split for this information and the training, or does information is "free"?

2. I am also confused by the sensitivity calculation, especially the factor k, can authors explain where does that come from? Isee they have used it from l2 norm, but please explain how is that equal to k.

3. I see no visible difference when epsilon is increased from 1-10-infinity, does PEARL's performance saturates at a privacy budget?

4. It will be nice to compare PEARL with other generative methods such as PATE-GAN, CTGAN for tabular data etc.


**Summary Of The Paper:**

Paper proposes a method for synthetic data generation called PEARL.

**Summary Of The Review:**

Paper proposes a new method (PEARL) to generate synthetic data based on CFs. It is a nice approach that seems to work.

---

> ### Author Response · Authors · 2021-11-22
> **Response to Reviewer**
>
> We thank the Reviewer for helpful comments. Please find the following our replies to the comments.
>
> 1) I am a bit confused by the auxiliary information release, is the total privacy budget split for this information and the training, or does information is "free"?
>
> As described in the beginning of Section 5.2 and footnote 3, the total privacy budget is split between releasing auxiliary information and the training. As the information's privacy is taken into account privacy-wise, it is not "free".
>
> 2) I am also confused by the sensitivity calculation, especially the factor k, can authors explain where does that come from? I see they have used it from l2 norm, but please explain how is that equal to k.
>
> We apologize for the lack of explanation there and a typo (a lack of square root; our implementation and results remained correct).
> The l2 norm of $\phi$ is taken over the k characteristic functions $\Phi$:
>
> $||{\phi}({x})\||_2 \equiv \sqrt{\sum_m |\Phi(t_m)|^2} \leq \sqrt{k}$
>
> where the sum is taken over $k$ characteristic functions and we have used $|\Phi(t_m)| \leq 1$. This is where the factor of $k$ comes from. A slightly more detailed description of the notations has been added at the end of Section 2.2
> Please let us know if there is anything else unclear.
>
> 3) I see no visible difference when epsilon is increased from 1-10-infinity, does PEARL's performance saturates at a privacy budget?
>
> Yes, the performance saturates at moderate values of epsilon. To understand this, we first note that at epsilon infinity, the performance saturates in the sense that the generated image quality is worse than those produced by GANs or VAEs. This is because of two reasons: 1) PEARL projects data to a lower dimension, causing loss of information even when no noise is added, 2) PEARL does not have a discriminator-like network to help train the generator. However, as the sensitivity is small (scales inversely proportional to the total sample size), the noise added (at epsilon 1-10) does not degrade the image quality much, except when the noise is extremely large (at small epsilon, e.g. 0.2). At these regions of epsilon, PEARL is able to produce better image quality than, e.g., DP-GAN. We mentioned this phenomenon in Section 5.2.
>
> 4) It will be nice to compare PEARL with other generative methods such as PATE-GAN, CTGAN for tabular data etc.
> We have added DPCGAN baselines for both tabular and image datasets during the rebuttal period.
> We did not use PATE-GAN as we have difficulty in replicating the result of PATE-GAN (as mentioned by other authors too, e.g., 2006.08265).
>
> DPCGAN is a conditional GAN and may be seen as a DP variant of CTGAN.
> We have also further evaluated another tabular dataset, Credit and added additional evaluation tasks for tabular data: marginal release, range query and MMD. We show in Appendix I that PEARL still performs favorably.

---

### Official Review · Reviewer_qSyM · 2021-11-03

**Correctness:** 4
**Technical Novelty And Significance:** 3
**Empirical Novelty And Significance:** 3
**Recommendation:** 8
**Confidence:** 4

**Main Review:**

The main strengths of the paper in my opinion are:
1) The approach is quite well motivated and the paper is well written.
2) The mathematical formulation is easy to follow and the approach is indeed quite novel
3) At least in the few empirical evaluations, this approach compares favorably against the other approaches compared against.

It seems to me, the main weakness of the paper is the lack of thorough benchmarking. For tabular data, it would be nice to see how the approach against PATE-GAN and on more than one dataset. For image data, it would be nice to see the benchmarking done on at least one somewhat more complex dataset (maybe CIFAR-10).

One minor comment is that the authors should consider moving algorithm 1 to the main paper.

**Summary Of The Paper:**

In this paper, the authors propose a novel differentially private approach to generate both continuous as well as discrete valued synthetic data. The authors utilize a one shot approach to providing the privacy, by first generating privatized embedding of the sensitive dataset. The privatized embeddings are then iteratively compared against the synthetic samples generated by the generator module using the characteristic function distance. The approach is similar to and can be considered a generalization of another new approach DP-MERF and seems to compare favorably in empirical experiments against DP-MERF and DP-GAN (two popular alternative approaches).

**Summary Of The Review:**

The paper presents a novel and well motivated approach to privacy preserving generative modeling. The approach has nice theoretical properties and compares well against other approaches on the few benchmark scenarios that the authors have looked at.

---

> ### Author Response · Authors · 2021-11-22
> **Response to Reviewer**
>
> We thank the Reviewer for helpful comments. Please find the following our replies to the comments.
>
> Q: it would be nice to see how the approach against PATE-GAN and on more than one dataset. For image data, CIFAR10 ...
>
> A: We have added DPCGAN baselines for both tabular and image datasets during the rebuttal period.
> We did not use PATE-GAN as we have difficulty in replicating the result of PATE-GAN (as mentioned by other authors too, e.g., 2006.08265). We did not evaluate CIFAR10 as it is difficult to generate complicated images like CIFAR10 under DP constraints; as far as we know, there is no state-of-the-art DP methods able to produce such images in the literature.
> We have also further evaluated another tabular dataset, Credit and added additional evaluation tasks for tabular data: marginal release, range query and MMD. We show in Appendix I that PEARL still performs favorably.
>
> Q: One minor comment is that the authors should consider moving algorithm 1 to the main paper.
>
> A: We give a high-level description of the algorithm in Section 5.1 instead as space constraints would not allow us to fit the algorithm to the main paper.

---

### Author Response · Authors · 2021-11-22
**To all reviewers**

We thank the reviewers for helpful comments. We have made changes in the main text based on the suggestions.
The major changes in our revised version is the addition of multiple baselines and datasets.
We also make minor changes in the text to improve clarity.
The detailed responses to the reviewers can be found in the following.

---

### Decision · Program_Chairs · 2022-01-20

**Decision:**

Accept (Poster)

**Comment:**

The paper presents a new framework of synthesizing differential private data using deep generative models. Reviewers liked the significance of the problem. They raised some concerns which was appropriately addressed in the rebuttal.  We hope the authors will take feedback into account and prepare a stronger camera ready version.